# Mitochondrial peptide BRAWNIN is essential for vertebrate respiratory complex III assembly

Shan Zhang[1], Boris Reljić [2,12], Chao Liang [1,12], Baptiste Kerouanton[1,12], Joel Celio Francisco [3], Jih Hou Peh[1], Camille Mary [4], Narendra Suhas Jagannathan [5], Volodimir Olexiouk [6], Claire Tang[1], Gio Fidelito [1], Srikanth Nama[7], Ruey-Kuang Cheng[8], Caroline Lei Wee [9], Loo Chien Wang[9], Paula Duek Roggli [10], Prabha Sampath [11], Lydie Lane [4], Enrico Petretto [5], Radoslaw M. Sobota [9], Suresh Jesuthasan [8,9], Lisa Tucker-Kellogg [3,5], Bruno Reversade [7,9], Gerben Menschaert[6], Lei Sun [1], David A. Stroud [2] & Lena Ho [1,7✉]

The emergence of small open reading frame (sORF)-encoded peptides (SEPs) is rapidly expanding the known proteome at the lower end of the size distribution. Here, we show that the mitochondrial proteome, particularly the respiratory chain, is enriched for small proteins. Using a prediction and validation pipeline for SEPs, we report the discovery of 16 endogenous nuclear encoded, mitochondrial-localized SEPs (mito-SEPs). Through functional prediction, proteomics, metabolomics and metabolic flux modeling, we demonstrate that BRAWNIN, a 71 a.a. peptide encoded by *C12orf73*, is essential for respiratory chain complex III (CIII) assembly. In human cells, BRAWNIN is induced by the energy-sensing AMPK pathway, and its depletion impairs mitochondrial ATP production. In zebrafish, *Brawnin* deletion causes complete CIII loss, resulting in severe growth retardation, lactic acidosis and early death. Our findings demonstrate that BRAWNIN is essential for vertebrate oxidative phosphorylation. We propose that mito-SEPs are an untapped resource for essential regulators of oxidative metabolism.

[1] Cardiovascular Metabolic Disorders Program, Duke-NUS Medical School, Singapore, Singapore. [2] Department of Biochemistry and Molecular Biology, The Bio21 Molecular Science & Biotechnology Institute, University of Melbourne, Melbourne, Australia. [3] Cancer and Stem Cell Biology Program, Duke-NUS Medical School, Singapore, Singapore. [4] SIB-Swiss Institute of Bioinformatics and Department of Microbiology and Molecular Medicine, Faculty of Medicine, Geneva University, Geneva, Switzerland. [5] Centre for Computational Biology, Duke-NUS Graduate Medical School Singapore, Singapore, Singapore. [6] Biobix, Lab of Bioinformatics and Computational Genomics, Department of Mathematical Modelling, Statistics and Bioinformatics, Ghent University, Ghent, Belgium. [7] Institute of Medical Biology, A*STAR, Singapore, Singapore. [8] Lee Kong Chian School of Medicine, Nanyang Technological University, Singapore, Singapore. [9] Institute of Molecular and Cell Biology, A*STAR, Singapore, Singapore. [10] SIB-Swiss Institute of Bioinformatics, Lausanne, Switzerland. [11] Skin Research Institute of Singapore, A*STAR, Singapore, Singapore. [12] These authors contributed equally: Boris Reljić, Chao Liang, Baptiste Kerouanton. ✉email: lena@ho-lab.org

The size and complexity of the human proteome is constantly evolving, ranging from genome-based estimations of 20,000 canonical proteins to over 70,000 if one considers protein variants (Uniprot, 2020). With the advent of ribosome profiling, the question of the size of the human proteome must once again be revisited. By detecting RNA translation at sub-codon resolution[1], ribosome profiling has unexpectedly uncovered thousands of coding regions hitherto thought to be translationally silent[2]. In particular, small open reading frames (sORFs) nested within long non-coding RNAs (lncRNAs), pseudogenes, intergenic transcripts or upstream of known ORFs (uORFs) that were previously not considered to be functional have now been shown to encode bona fide peptides[2–5]. Evidence accumulating within the last decade point to the widespread prevalence of sORF-encoded peptides, or SEPs, and have spurred multiple independent efforts to discover and catalog them[6–8]. From a biological point of view, the SEP mini-proteome is a rich repository of unexplored gene functions. However, unlike the high throughput techniques that enabled their discovery, functional de-orphanization of SEPs remains a resource-intensive endeavor that relies on fundamental approaches in experimental biology. To date, SEPs have been implicated in diverse functions ranging from cardiovascular and placental development, mTORC1 activation, calcium signaling, mRNA de-capping to membrane fusion, and oocyte fertilization[9–15]. However, whether SEPs have programmatic functions enabled by their small size remains an open question. To this end, we asked if SEPs were particularly enriched in subcellular locations that would provide instructive cues for their functional characterization.

## Results

**Mitochondria proteome is enriched in SEPs**. To this point, we observed that the mitochondria proteome, particularly at the inner mitochondrial membrane (IMM) (cataloged in Mitocarta 2.0[16]) is enriched for proteins smaller than 100 a.a. i.e. SEPs ($p$-value = $2.2 \times 10^{-16}$) (Fig. 1a–c), and speculated that mitochondria are a hotspot for SEP function. To uncover novel mitochondria-targeted SEPs (mito-SEPs) from the human SEP peptidome[7], 2343 high-quality human SEPs encoded by annotated short-coding sequence (CDS) genes, lncRNAs, uORFs, and intergenic ORFs were selected for test of mitochondrial localization (Fig. 1d, details in Supplementary Note 1). We first employed a three-prong strategy that identified a candidate mito-SEP on the basis of a mitochondrial gene expression signature (Supplementary Fig. 1a); a mitochondrial targeting protein domain (Supplementary Fig. 1b) or empirical detection of the SEP in mass spectrometric data of purified mitochondria (details in Supplementary Note 1). Altogether, our mito-SEP discovery pipeline identified 173 candidates (Fig. 1d and Supplementary Data 1), including 2 positive controls MKKS uORF1 and uORF2[17]. HA-tagged mito-SEP candidates were experimentally validated by immunofluorescence of transfected HeLa cells (Fig. 1e). In total, we successfully validated protein expression for 88/173 sORFs (Supplementary Data 1), defined as the ability of the ORF to generate a peptide with sufficient stability to enable detection (Supplementary Fig. 1c–e). Of these, 23% (20/88) of successfully expressed peptides displayed unambiguous mitochondrial localization including the MKKS uORFs (Fig. 1e). Among the positive candidates were MTLN/MOXI and MIEF1 uORF which were subsequently independently discovered to have critical roles in mitochondria metabolism and mtDNA translation[18–21], providing proof for the robustness of our pipeline. All in all, our screen identified 16 bona fide mito-SEPs of unknown function (MSUF) encoded by uORFs, lincRNAs and short CDSs (Fig. 1e and Table 1).

**Preferential use of SEPs in respiratory complexes**. Mitochondria are the principal sites of energy conversion where ATP is generated through the process of oxidative phosphorylation (OXPHOS). Defects in OXPHOS are responsible for many inherited mitochondrial diseases and manifest in aging and metabolic disorders such as diabetes, heart disease, and cancer[22]. Notably, 63.5% (40/63) of known mitochondrial low molecular weight (MW) proteins function as assembly factors or core subunits of respiratory chain (RC) complexes that carry out OXPHOS (Supplementary Fig. 1f). RC complexes are vice versa enriched for low MW proteins (Supplementary Fig. 1g). Of the 16 mito-SEPs, the expression of 8 correlated with respiratory chain and electron transport function above the 90th percentile (Fig. 1f), suggesting that half of them participate directly or indirectly in oxidative phosphorylation.

**BRAWNIN resides in the IMM and responds to AMPK**. We selected a mito-SEP encoded by the *C12orf73* gene, which we renamed *BRAWNIN (BR)* (Fig. 2a and Supplementary Fig. 2a), for further characterization because its stable overexpression in U87MG significantly enhanced OXPHOS (Supplementary Fig. 2b). The BR peptide is conserved in vertebrates (Fig. 2b) but not in yeast and nematodes. Although ribosome profiling detected 2 plausible BR peptides arising from alternative splicing (Fig. 2a and Supplementary Fig. 2c), we confirmed that only the cDNA of the long isoform (P1) produced a stable 8-kDa peptide, while the short isoform (P2) was remarkably less stable even when over-expressed (Supplementary Fig. 2d). Endogenous BR was detected by western blotting with a custom α-BR antibody in HEK293T and HeLa cell lysates (Fig. 2c) and can be depleted by siRNAs targeting the *BR* transcript (Fig. 2d, e). Endogenous BR is enriched in mitochondria-rich fractions (Fig. 2f), and co-localized with Mitotracker in both HeLa and HEK293T (Fig. 2g, h). Altogether, these data confirm that BR is a nuclear-encoded peptide that is imported into the mitochondria. In vivo, Br can be detected in mouse skeletal muscle where it co-localizes with matrix marker Citrate Synthase (CS) and with a mitochondria-targeted GFP transgene mito-Dendra2[23] (Fig. 2i, j). BR is also detectable in human cardiac and skeletal muscle (Supplementary Fig. 2e) where it displays a staining pattern characteristic of the mitochondria network. Moreover, BR protein abundance in mouse tissues correlated well with that of mitochondrial respiratory chain proteins, being high in brown adipose, cardiac and skeletal muscle but virtually undetectable in white adipose tissue (Supplementary Fig. 2f). Together, these results establish that BR is a bona fide endogenous mito-SEP both in vitro and in vivo.

Because mitochondria are highly compartmentalized, the molecular function of a protein determines its sub-mitochondrial location and vice versa[24]. We first established that BR is membrane-bound by sodium carbonate extraction (Fig. 2k). Next, using differential detergent extraction, we determined that BR resides in the IMM and not in the outer mitochondrial membrane (OMM) (Fig. 2l). Proteinase K protection assays further suggest that the C-terminus of BR faces the intermembrane space (IMS) (Fig. 2m). In contrast to its clear mitochondrial localization, human BR lacks a classical mitochondrial targeting sequence (MTS); its N-terminus is instead predicted to function as a secretory signal peptide by SignalP[25]. Indeed, when overexpressed, BR can be detected in the supernatant (Supplementary Fig. 2g). However, this may not represent bona fide secretion as it cannot be inhibited by brefeldin A. Hence, the N-terminal "signal peptide", which is not cleaved (Supplementary Fig. 2h), functions as a mitochondrial targeting single-pass transmembrane domain (TMD) to anchor endogenous BR into

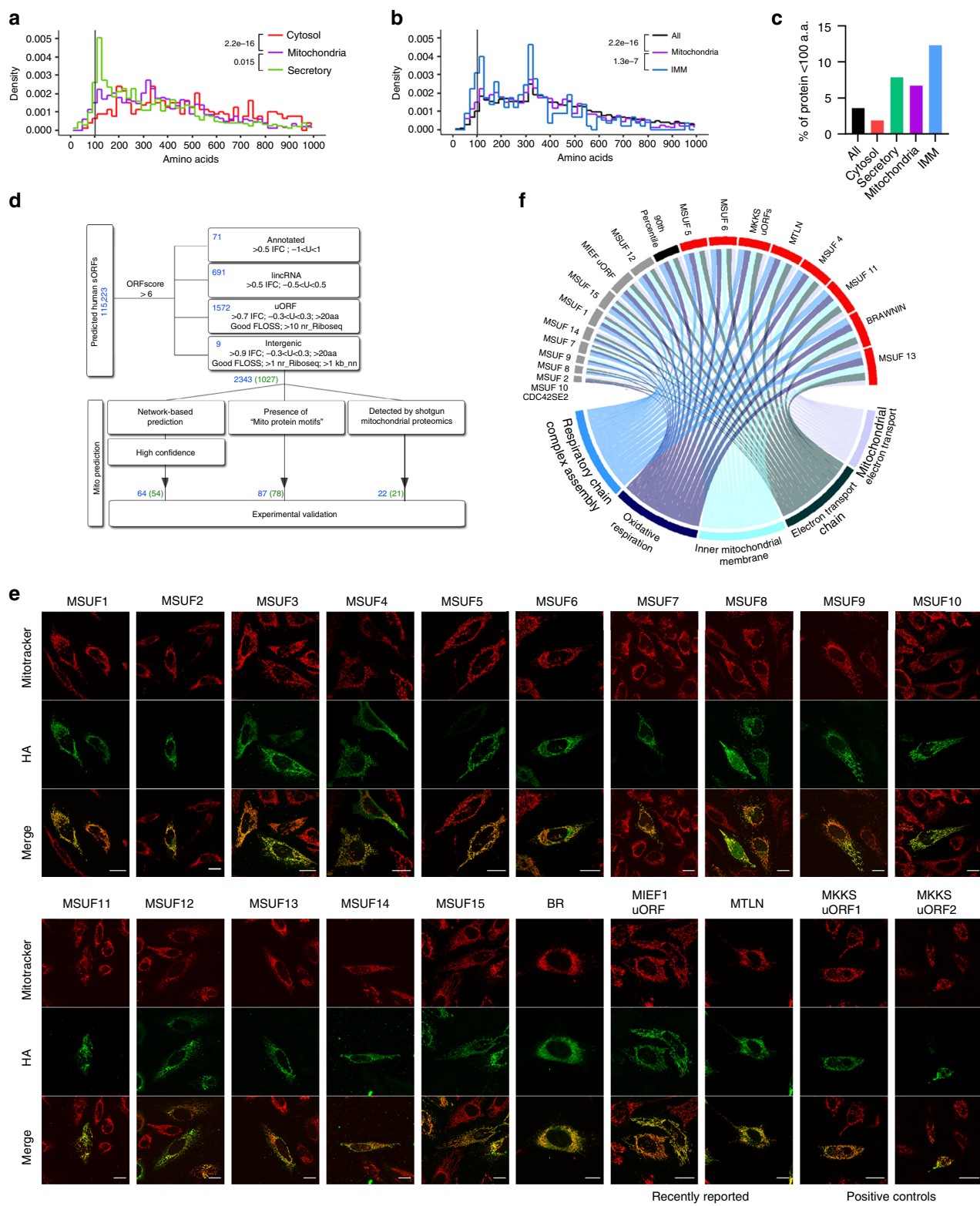

the IMM. Consistent with this, BR-P2, which contains the full TMD of BR-P1, is similarly targeted to the mitochondria despite its relative instability (Supplementary Fig. 2i). Removal of the predicted TMD from BR (BR$^{\Delta N25}$), while causing its near complete destabilization (Supplementary Fig. 2j), targets BR$^{\Delta N25}$ to the cytosol instead of the mitochondria (Supplementary Fig. 2k).

Proteins in the IMM participate in a wide variety of processes, most commonly OXPHOS, calcium import, and the transport of proteins and metabolites. Co-expression analysis by weighted gene co-expression network analysis (WGCNA)[26] supports the notion that BR participates in electron transport and OXPHOS in multiple human organs and species (Fig. 3a, Supplementary Fig. 3a, Supplementary Data 2). Components of the respiratory

**Fig. 1 Mito-SEP prediction pipeline identifies uncharacterized endogenous mitochondrial SEPs. a** Size distribution of curated human proteins annotated as mitochondria (1172 proteins), cytosolic (5806 proteins), or secretory (3351 proteins) on Uniprot. *p*-value by two-sided Mann–Whitney *U*-test. n.s. = not significant. **b** Size distribution of Uniprot proteins (All) (20417 proteins) annotated to reside in mitochondria (1172 proteins) or inner mitochondrial membrane (IMM) (292 proteins). *p*-value by two-sided Mann–Whitney U-test. **c** Percentage of proteins smaller than 100 a.a. in different cellular localizations. **d** Human SEP selection and mitochondrial prediction workflow. Blue numbers indicate number of SEPs and green numbers indicate number of genes encoding these SEPs. **e** HA immunofluorescence (green) of mito-SEP candidates showing colocalization with Mitotracker Red (red) in HeLa cells. Uncharacterized SEPs are labeled as mitochondrial SEP with unknown function (MSUF). Scale = 20 μM. **f** Circular plot of expression correlation between MSUF and MSigDB gene ontology (GO) gene sets that relate to oxidative phosphorylation. The width of each connector represents the normalized enrichment score (NES) value of the gene set enrichment analysis (GSEA) analysis performed on a ranked list of MSUF-coexpressed genes determined by weighted gene co-expression network analysis (WGCNA). The black bar indicates the score of the 90th percentile of the score of 1000 randomly selected genes. *CDC42SE2* depicts the position of a non-mitochondrial negative control with NES = 0.

**Table 1 List of verified mito-SEPs.**

| Label | Name | Gene ID | Locus | Motif | Length (a.a.) |
|---|---|---|---|---|---|
| BRAWNIN | BRAWNIN | ENSG00000204954 | sORF | SP | 71 |
| MSUF1 | PIGBOS1 | ENSG00000225973 | sORF | TMD | 54 |
| MSUF2 | LINC01560 | ENSG00000196741 | lncrna | Not detected | 94 |
| MSUF3 | LINC00473 | ENSG00000223414 | lncrna | TMD | 90 |
| MSUF4 | SMIM12 | ENSG00000163866 | sORF | TMD | 92 |
| MSUF5 | SMIM26 | ENSG00000232388 | sORF | TMD | 95 |
| MSUF6 | C16ORF91 | ENSG00000174109 | sORF | TMD | 65 |
| MSUF7 | PRPF19 uORF | ENSG00000110107 | 5UTR | MTS | 66 |
| MSUF8 | UBQLN1 uORF | ENSG00000135018 | 5UTR | MTS | 53 |
| MSUF9 | RNASEH2C uORF | ENSG00000172922 | 5UTR | MTS | 79 |
| MSUF10 | CENPW uORF | ENSG00000203760 | 5UTR | MTS | 68 |
| MSUF11 | MRPL24 uORF | ENSG00000143314 | 5UTR | Not detected | 77 |
| MSUF12 | TMEM222 uORF | ENSG00000186501 | 5UTR | TMD | 90 |
| MSUF13 | EIF3K uORF | ENSG00000178982 | 5UTR | Not detected | 59 |
| MSUF14 | KIAA0100 uORF | ENSG00000007202 | 5UTR | TMD | 73 |
| MSUF15 | SLC35A4 uORF | ENSG00000176087 | 5UTR | TMD | 96 |
| MTLN | LINC00116 | ENSG00000175701 | sORF | TMD | 56 |
| MIEF1 uORF | MIEF1 uORF | ENSG00000285025 | 5UTR | MTS | 70 |

*SP* signal peptide, *TMD* transmembrane domain, *MTS* mitochondrial targeting sequence.

chain are dynamically regulated by energy status. In response to a reduction of cellular ATP:ADP ratio, such as during nutrient depletion or exercise, the energy-sensing AMP-activated protein kinase (AMPK) pathway restores energetic balance by stimulating ATP-producing catabolic pathways and inhibiting energy-consuming anabolic processes[27]. As such, we hypothesized that BR levels would be regulated by cellular energy status. Indeed, activation of AMPK using 5-aminoimidizole-4-carboxamide-1-β-D-riboside (AICAR)[28] robustly increased protein levels of BR (Fig. 3b). Likewise, BR increased in response to glucose, serum and fatty acid starvation, in parallel with AMPK activation marked by phosphorylation of Acetyl-CoA carboxylase (ACC), a direct AMPK target[29] (Fig. 3c). In mouse myotubes, enforced expression of PGC-1α, the AMPK-activated regulator of mitochondrial biogenesis[30], robustly induced *Br* expression along with other OXPHOS components (Supplementary Fig. 3b). These data together indicate that BR is under the regulatory control of the AMPK- PGC-1α energy homeostasis axis, and predict that a loss of BR would impair cellular bioenergetics and mitochondrial ATP production. To investigate this, we silenced *BR* in U87MG glioblastoma cells (a highly oxidative cell type) using both transient siRNA and stable shRNA-mediated approaches. Indeed, BR depletion significantly decreased basal, maximal respiration, spare capacity, and reduced ATP production in both si*BR* and sh*BR* U87MG (Fig. 3d–f, Supplementary Fig. 3c). Likewise, HEK293T grown in galactose displayed the same reduction in basal respiration (Supplementary Fig. 3d), which sufficiently perturbed cellular ATP:ADP ratio to activate AMPK (Supplementary Fig. 3e). These data confirm that *BR* is essential for

cellular bioenergetics, potentially through regulation of the electron transport chain.

**Deletion of *Br* in zebrafish causes mitochondrial disease.** *BR* is highly conserved in vertebrates, suggesting that its role in bioenergetics is fundamental and required throughout evolution. To understand the in vivo requirement of BR in animal physiology, we turned to the zebrafish where *br* is most highly expressed in skeletal and cardiac muscle (Fig. 4a). Using Crispr/Cas9, we generated protein null *br* knockout animals (KO) by removing the entire protein-coding ORF contained in exons 2 and 3 (Supplementary Fig. 4a and Fig. 4b). *br* zygotic KO (zKO) larvae were produced at Mendelian ratios (Fig. 4c) with no gross development defects within the first 5 days besides a delay in swim bladder inflation (Supplementary Fig. 4b, c). Muscle formation was normal with no overt signs of myopathy or dystrophy (Supplementary Fig. 4d). However, if left unsegregated, *br* zKOs could not be recovered by 65 days post fertilization (dpf) (Fig. 4c). We therefore segregated larvae from intercrosses at 3 dpf and allowed them to grow at low density. Under this regime, zKO juveniles survived into adulthood, but displayed growth retardation from 28 dpf (Fig. 4d) that resulted in large differences in body size by 65 dpf (Fig. 4e). zKOs are sub-fertile, and can generate maternal zygotic knockout embryos (mzKO) albeit at a very low frequency (~1/20 mating pairs). Because Br is a mitochondrial peptide, it is maternally deposited in oocytes. mzKOs therefore lack any residual Br protein that is available to zKOs during the initial phases of larvagenesis. Consequently, *mzKO*

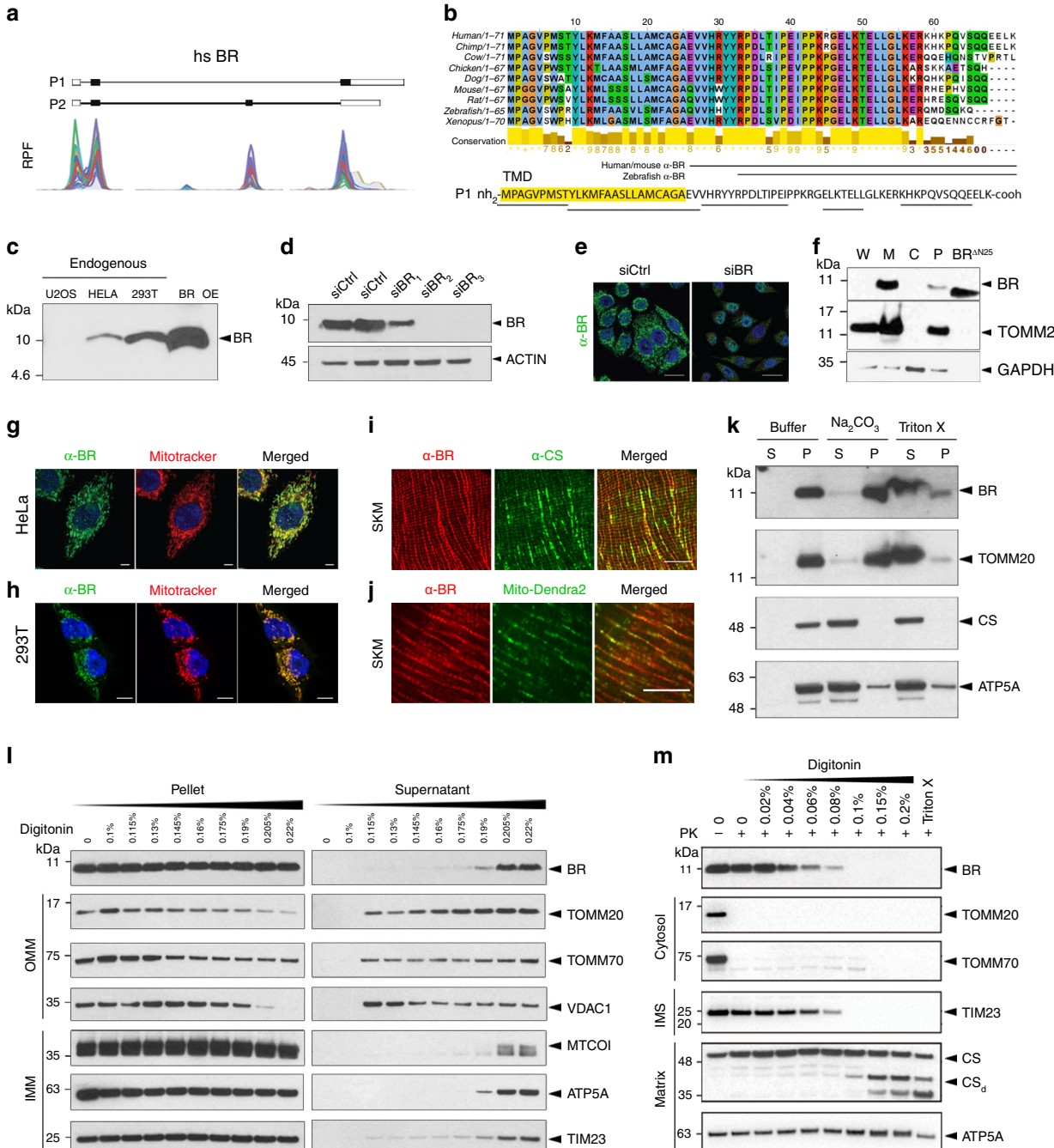

**Fig. 2 BRAWNIN (BR) is a conserved SEP at the inner mitochondrial membrane. a** Ribosome protected fragment (RPF) reads from the *BRAWNIN* gene (*C12orf73*) in human cell lines. P1 and P2 refer to the 2 potential ORF isoforms generated by alternative splicing. **b** BR (P1) peptide is conserved throughout vertebrate evolution and possesses an N-terminal hydrophobic region that is predicted to be a transmembrane domain or signal peptide. Underlined regions are peptides detected by mass spec. Regions used as immunogens for raising the polyclonal antibodies used in this study are indicated. **c** Endogenous BR is detected in HeLa and HEK293T with an α-BR antibody. OE = overexpression. **d** Western blot of HEK293T lysates transfected with control or *BR*-targeting siRNAs. **e** Immunofluorescence of endogenous BR detected by α-BR in HeLa transfected with control or *BR*-targeting siRNAs. Scale = 5 μM. **f** Western blot of indicated proteins from subcellular fractions of HEK293T. W = whole cell, M = mitochondria enriched fraction, C = cytosol, P = nuclear fraction and debris, including un-extracted mitochondria; $BR^{\Delta N25}$ = synthetic BR peptide lacking the N-terminal 25 a.a. **g** Endogenous BR detected by α-BR in HeLa costained with mitotracker. Scale = 5 μM. **h** Endogenous BR detected by α-BR in HEK293T costained with mitotracker. Scale = 5 μM. **i** α-BR and α-citrate synthase (CS) immunofluorescence in mouse skeletal muscle, longitudinal section. Scale = 20 μM. **j** α-BR immunofluorescence and GFP epifluorescence of Mito-Dendra2 in mouse skeletal muscle, longitudinal section. Scale = 20 μM. **k** Extraction of endogenous BR from HEK293T mitochondria with buffer alone, $Na_2CO_3$ solution or buffer containing 1% Triton X. S = supernatant; P = pellet. **l** Extraction of endogenous BR from HEK293T mitochondria using increasing concentrations of digitonin. TOMM20, TOMM70, and VDAC1 are outer mitochondrial membrane (OMM) proteins, are MTCO1, ATP5A, and TIM23 are IMM associated. **m** Proteinase K protection assay of HEK293T mitochondria. The α-BR epitope is C-terminal to the predicted transmembrane domain. TOMM20 and TOMM70 have cytosolic domains. TIM23 has IMS domains. Both CS and ATP5A are in the matrix.

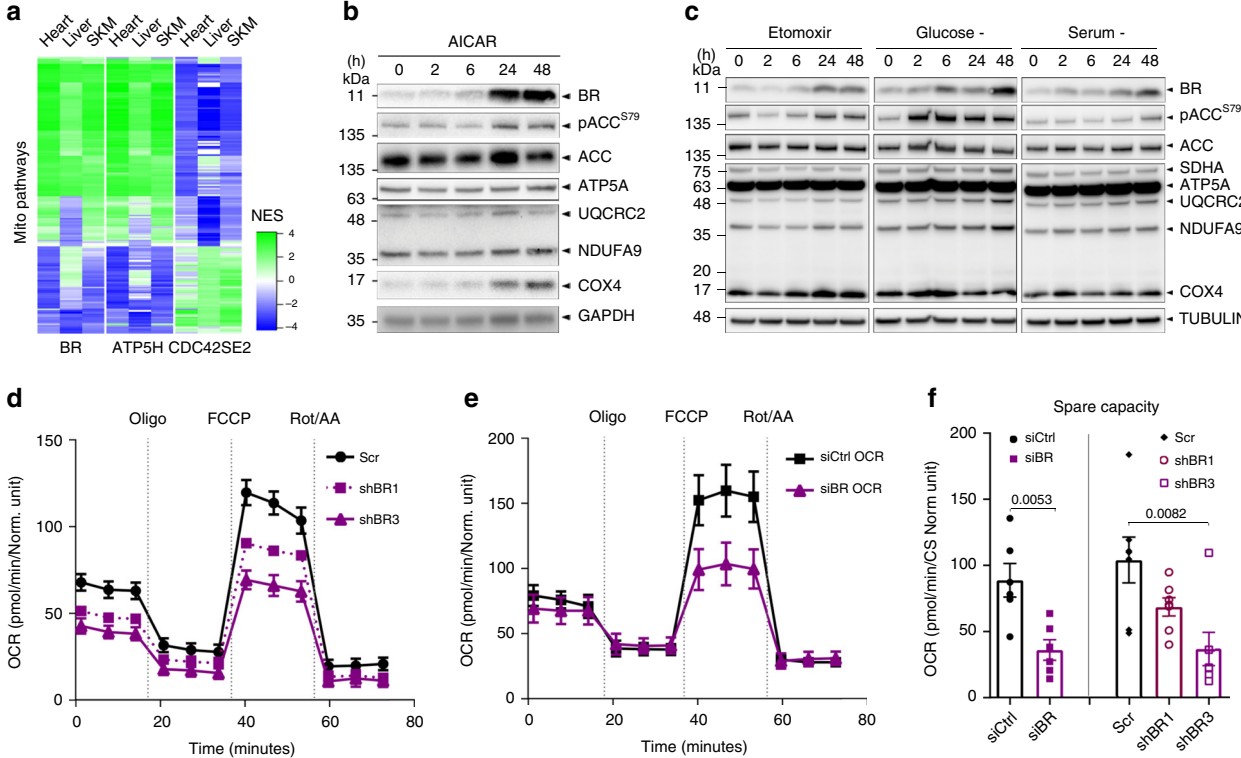

**Fig. 3 BR is an AMPK target that potentiates OXPHOS in human U87MG cells. a** Heatmap depicting GSEA NES scores of mitochondrial pathways (Supplementary Data 2) in the ranked list of pairwise correlations between genes in the human liver, heart, or skeletal muscle (SKM) with either *BR* gene, known electron transport gene *ATP5H* or *CDC42SE2*, a representative non-mitochondrial candidate SEP. **b** SDS-PAGE of HEK293T treated with AMPK activator AICAR (500 µM) for the indicated time periods. **c** SDS-PAGE of HEK293T treated with indicated starvation regimes for the indicated time periods. Etomoxir was used at 3 µM. **d** Mitostress test (Agilent Seahorse) analysis of control versus stable *BR* knockdown U87MG cells. Oxygen consumption rate (OCR) reads were normalized with citric synthase (CS) activity. Data are mean and SEM of six technical replicates. Experiment is representative of two biological replicates. Oligo = oligomycin, FCCP = carbonyl cyanide 4-(trifluoromethoxy)phenylhydrazone, Rot/AA = rotenone/ antimycin A. **e** Analogous to d. except using short-term (72 h) siRNA-mediated depletion of BR. The experiment is representative of three biological replicates each with six technical replicates. Error bars represent SEM. **f** Spare respiratory capacity calculated from experiments in **d** and **e**. Data are mean and SEM of six technical replicates which are representative of two and three biological replicates for shRNA and siRNA-mediated depletion, respectively. *p*-values from two-sided unpaired *t*-test.

had a more severe phenotype compared to *z*KOs (Fig. 4d). By 6 dpf, *mz*KO larvae were visibly less active despite fully inflated swim bladders (Supplementary Fig. 4e). By 7 dpf, at the onset of feeding, their food intake was significantly less than WT larvae (Supplementary Fig. 4f). By 11 dpf, they displayed a dramatic reduction of directional swimming and distance swam compared to WTs (Fig. 4f and Supplementary Fig. 4g). Consequentially, *mz*KO larvae did not increase in body length even under low density segregated growth conditions, and did not survive past 42 dpf under the conditions of our aquatics facility (Fig. 4d). In line with our observations in human U87MG cells, basal, ADP stimulated, maximal respiration, and ATP production in *br* KO skeletal muscle mitochondria were reduced to half that of WT (Fig. 4g). This in turn significantly reduced whole animal respiration *br* mz KO compared to WT larvae even as early as 1 dpf (Fig. 4h), leading to a failure to thrive and early death (Fig. 4d). These data provide evidence that BR is mito-SEP with conserved and critical functions in mediating OXPHOS, and in vivo deletion of *br* causes severe mitochondrial deficiency leading to growth retardation.

**BRAWNIN is required for respiratory chain complex III.** To understand the molecular basis of the mitochondrial defect in *br* KO fish, we profiled the metabolomic perturbations in whole *mz*KO larvae at 5 dpf. Targeted metabolomics analysis revealed a

2-fold increase in the levels of lactate in *mz*KO larvae (Fig. 5a) and a 3-fold increase in the citric acid cycle intermediate succinate as compared to WT larvae (Fig. 5b). Lactate accumulation is caused by decreased pyruvate oxidation secondary to mitochondrial deficiency and is frequently used as a diagnostic symptom for patients with mitochondrial diseases[31]. To understand what molecular defects in the mitochondria could result in these perturbations, we turned to MitoCore, a metabolic flux model that simulates over 300 mitochondrial reactions in a human cardiomyocyte using flux balance analysis[32]. By blocking each reaction in turn, and simulating 5000 feasible flux states for each possible blockage, we found that inhibiting ETC Complex III (CIII, Fig. 5c and Supplementary Fig. 5a) and Complex IV (CIV, Supplementary Fig. 5b) flux caused the largest, most biologically plausible increase in succinate and lactate (as inferred from the export rate of the two metabolites). Similarly, when we deleted each of the 371 genes in the MitoCore model, the deletion of genes encoding CIII and CIV components caused the largest increase in succinate and lactate (Supplementary Table 1). In contrast, inhibiting Complex II was not sufficient to induce the observed elevations in lactate (Supplementary Fig. 5c). These data suggest that the underlying molecular defect in *br* KO mitochondria is related to CIII and/or CIV dysfunction.

To investigate this possibility, we constructed the BR interactome in HEK293T mitochondria by co-immunoprecipitating

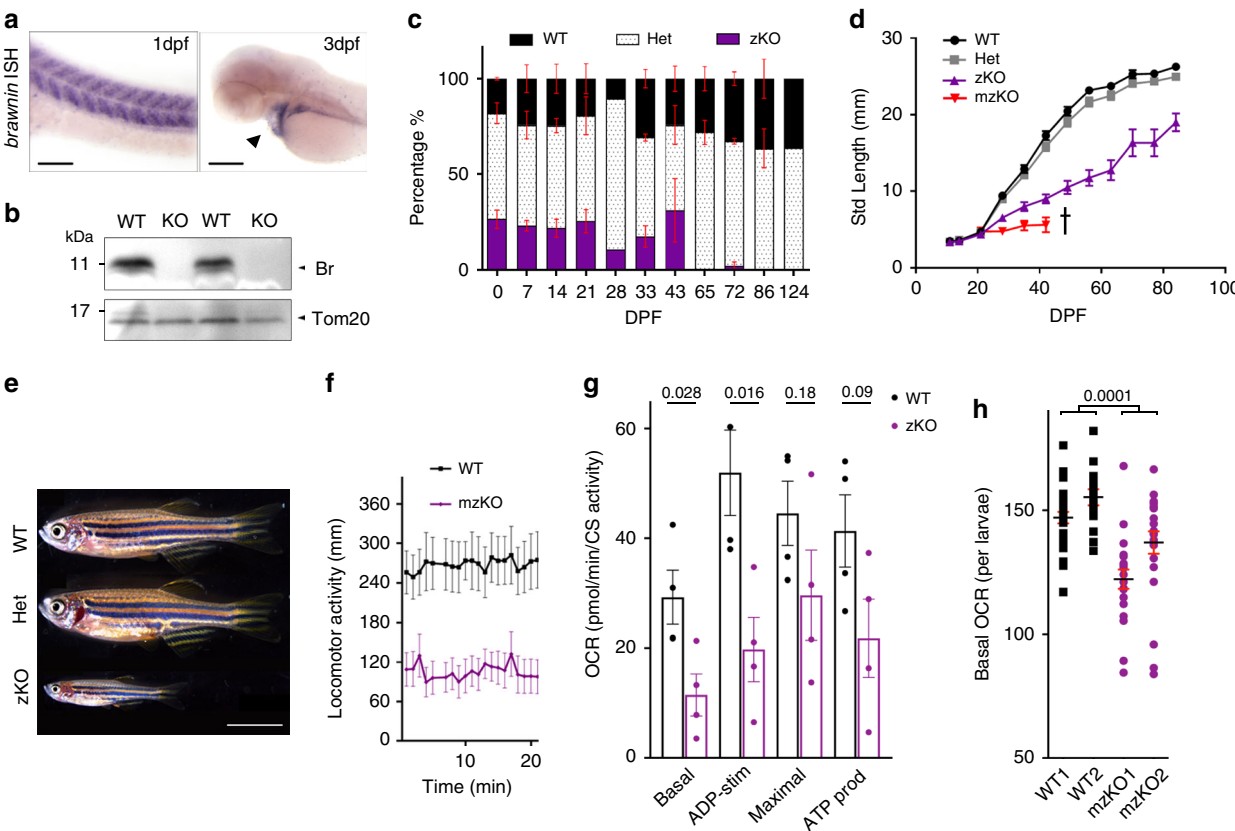

**Fig. 4 Knockout of *Br* in zebrafish causes lethal mitochondrial deficiency. a** In-situ hybridization (ISH) of *brawnin (br)* in zebrafish larvae. dpf = days post fertilization. Left panel: scale = 100 μm, right panel: scale 200 μm. **b** Western blot analysis of purified mitochondria from adult WT and KO (homozygous $br^\Delta/br^\Delta$) skeletal muscle using zebrafish-specific α-Br. **c** Mendelian percentages of offspring from heterozygous (Het) intercrosses genotyped at the indicated dpf. Data represent the mean of at least 2 clutches per time point (except 28 and 124 dpf) with at least 15 animals per clutch. Error bars indicate SEM. **d** Standard length (mm) measurements of WT, Het, and zygotic KO (zKO) larvae generated from an F2 intercross genotyped and segregated at 3 dpf. Maternal zygotic KOs (mzKOs) were generated from a homozygous incross and measured until 42 dpf when all animals succumbed to disease. Data are representative of four independent experiments from two independent Cas9/gRNA injected $br^\Delta$ KO allele founders. Number of animals at the beginning of the experiment: WT = 14; HET = 37; zKO = 8; mzKO = 22. Error bars indicate SEM. **e** Representative images of WT, Het, and zKO clutch-mates at 65 dpf. Scale = 5 mM. **f** Basal motility monitoring continuous swim tracking of WT and mzKO larvae at 11 dpf. 23 animals per genotype monitored. Error bars indicate SEM. **g** Coupled respiratory analysis of isolated SKM mitochondria measured on Agilent Seahorse in the presence of succinate + rotenone, read out as CS-normalized OCR. Basal, ADP stimulated, maximal respiration and ATP production were plotted. Data represent mean and SEM of four biological replicates. *p*-values from two-sided paired *t*-test. **h** Basal respiratory rate of free moving 1 dpf WT and mzKO larvae measured on Agilent Seahorse platform. Number of animals: WT1 = 30; WT2 = 15; KO1 = 21; KO2 = 23. Data represent mean and SEM, *p*-values from two-sided unpaired *t*-test.

either endogenous or overexpressed BR followed by mass spectrometry (IP/MS). At the intersection of 4 independent IP/MS experiments were three proteins: BR, GHITM, and UQCRC1 (Fig. 5d, Supplementary Data 3). UQCRC1 is a core component of CIII[33] while GHITM has been proposed to regulate cytochrome c release during apoptosis[34]. The interaction with GHITM could not be validated with several commercially available antibodies. In contrast, the interaction between BR and UQCRC1 detected in human cells was validated by co-immunoprecipitation(co-IP)/western blot in digitonin-solubilized mouse heart mitochondria expressing a Br-FLAG construct (Fig. 5e). Likewise, BR was co-immunoprecipitated by CIII subunit UQCRQ using a surface exposed C-terminal FLAG tag in HEK293T (Fig. 5f). These data suggest that BR, an IMM peptide, directly interacts with the respiratory chain at CIII.

Among all genes encoding RC proteins, *BR* transcript expression correlates most strongly with those encoding CIII assembly factors (Supplementary Data 4). On this basis, we hypothesized that the observed respiratory defects in si/sh*BR* U87MG and *br* KO fish are caused by CIII deficiency. To address

this possibility, we performed quantitative proteomics on isolated skeletal muscle mitochondria from *br* WT and KO animals (Fig. 6a). Strikingly, we found that 5 subunits of CIII were significantly downregulated, while subunits of CI, II and IV were not significantly affected (Fig. 6b, c, Supplementary Data 5), suggesting that CIII stability or assembly is specifically impaired in the absence of br. Indeed, SDS-PAGE analysis of *br* KO mitochondria confirmed that total Uqcrc1 and Uqcrc2 levels were reduced by >80% (Fig. 6d, e) resulting in a >90 % reduction of mature CIII dimers (CIII₂) as measured by Uqcrc1, Uqcrc2, Uqcrh and Uqcrb densitometry in blue-native (BN)-PAGE (Fig. 6f). The specific reduction in CIII₂ was also seen in sh*BR* U87MG cells both at steady state (Supplementary Fig. 6a–c) and during active assembly of respiratory chain complexes released from doxycycline-mediated inhibition of mitochondrial translation (Supplementary Fig. 6d). Furthermore, deletion of murine *Br* in mouse embryonic fibroblasts (MEF) (Supplementary Fig. 6e, f) likewise resulted in reduction of CIII subunit proteins levels and assembled CIII₂ (Supplementary Fig. 6g, h), demonstrating that the requirement for BR in CIII assembly or stability extends to

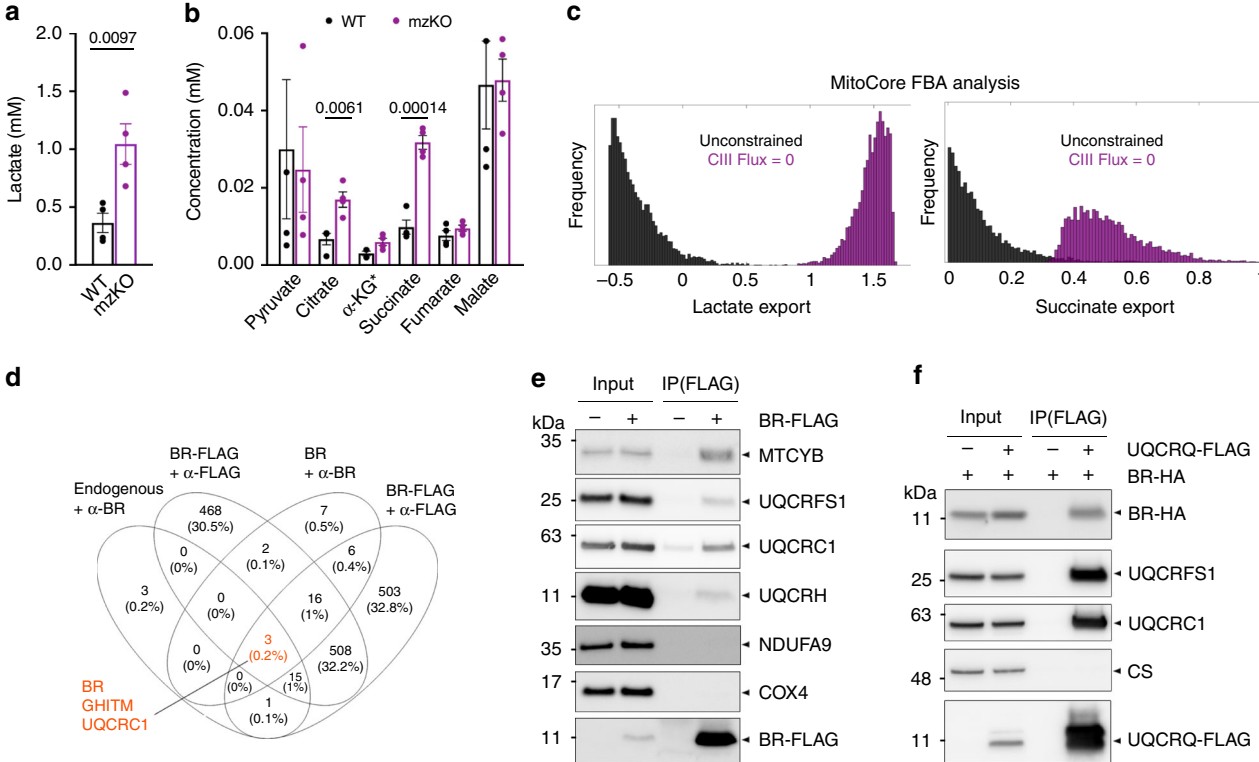

**Fig. 5 Br functionally and physically interacts with electron transport chain complex III. a** Lactate concentrations in 5 dpf WT and mzKO larvae. Data represent mean and SEM of four independent clutches each with 50 larvae per genotype. *p*-values from two-sided paired *t*-test. **b** Citric acid cycle organic acid concentrations in 5 dpf WT and mzKO larvae. Data represent mean and SEM of four independent clutches each with 50 larvae per genotype. *p*-values from two-sided paired *t*-test with Bonferroni correction. *α-KG concentrations are a 10-fold multiple to enable visualization on the same scale. **c** Distributions of the export rate of lactate and succinate (5000 flux samples) comparing the unconstrained MitoCore model versus the CIII-blocked model derived by MitoCore flux balance analysis (FBA). **d** Venn diagram depicting intersections of proteins identified in 4 co-IP/mass spec experiments of BR in HEK293T mitochondria. Endogenous BR and overexpressed untagged BR were immunoprecipitated with α-BR, overexpressed BR-FLAG with α-FLAG. **e** Immunoprecipitation of BR-FLAG followed by immunoblotting for indicated ETC proteins from mouse mitochondria with AAV9-mediated expression of BR-FLAG. MTCYB, UQCRFS1, UQCRC1, and UQCRH are components of CIII. NDUFA9 and COX4 are components of CI and CIV, respectively. **f**. Reciprocal IP of BR with CIII components. UQCRQ-FLAG and BR-HA were co-transfected into HEK293T. UQCRQ-FLAG immunoprecipitation was followed by immunoblotting for BR-HA and indicated proteins. CS is an unrelated matrix protein.

higher vertebrates. Furthermore, *Br* KO MEFs activated the AMPK pathway due to the energetic perturbation caused by CIII deficiency (Supplementary Fig. 6i). Finally, the observed reductions in CIII protein levels led to significant reduction of all CIII activity as measured by spectrophotometric measurement of RC enzymatic activities[35] (Fig. 6g), while CI, CII, and CIV activities were not significantly affected. Altogether, our data demonstrate that the loss of *Br* in zebrafish causes overt and lethal mitochondrial deficiency due to the specific loss of CIII in the respiratory chain.

## Discussion

Our findings in this report demonstrate that SEPs are preferentially enriched in the mitochondria and play programmatic functions in oxidative respiration. We speculate that this size bias might simply reflect the endosymbiotic origins of the mitochondrion. However, since the mito-SEPs including BR found in this study are not conserved in prokaryotes, an alternative explanation lies in the energetic barrier imposed by the double membranes of the mitochondrion against protein import. Simply put, it is energetically less costly to import small proteins than large proteins. For an organelle that is primarily responsible for maintaining energy balance, such a strategy would allow the

mitochondrion to lower the cost of its own biogenesis during a time of increased ATP demand, allowing homeostasis to be reached in a shorter time frame.

The mechanism with which BR promotes CIII assembly and/or stability also calls for greater in-depth investigation. The assembly of complex III in mammals is poorly understood and largely inferred from studies performed in yeast[36]. In zebrafish, loss of Br caused complete attenuation of the fully assembled 500-kDa CIII$_2$, with no evidence of immature pre-CIII$_2$ assemblies seen in mutants of late assembly factors[37–39], arguing that Br acts early in the biogenesis and known sequence of CIII assembly before dimerization of assembly intermediates[37]. Consistent with a possible direct role of Br as a CIII assembly factor in vivo, BN- and SDS-PAGE demonstrated partial co-migration of endogenous Br with CIII$_2$ in zebrafish mitochondria (Supplementary Fig. 6j, k). However, our data currently cannot exclude the possibility that BR mediates CIII biogenesis by facilitating the import of its constituent subunits or their stability following translation/import, which will require further investigation. BR is highly responsive to energy status. Electrons transferred from the oxidation of divergent fuel types converge at CIII. Because CIII is the center of the respiratory chain, we speculate that BR, acting as an early CIII assembly factor, coordinates the bioenergetic output of the respiratory chain with nutrient availability to maintain energy homeostasis. Conversely,

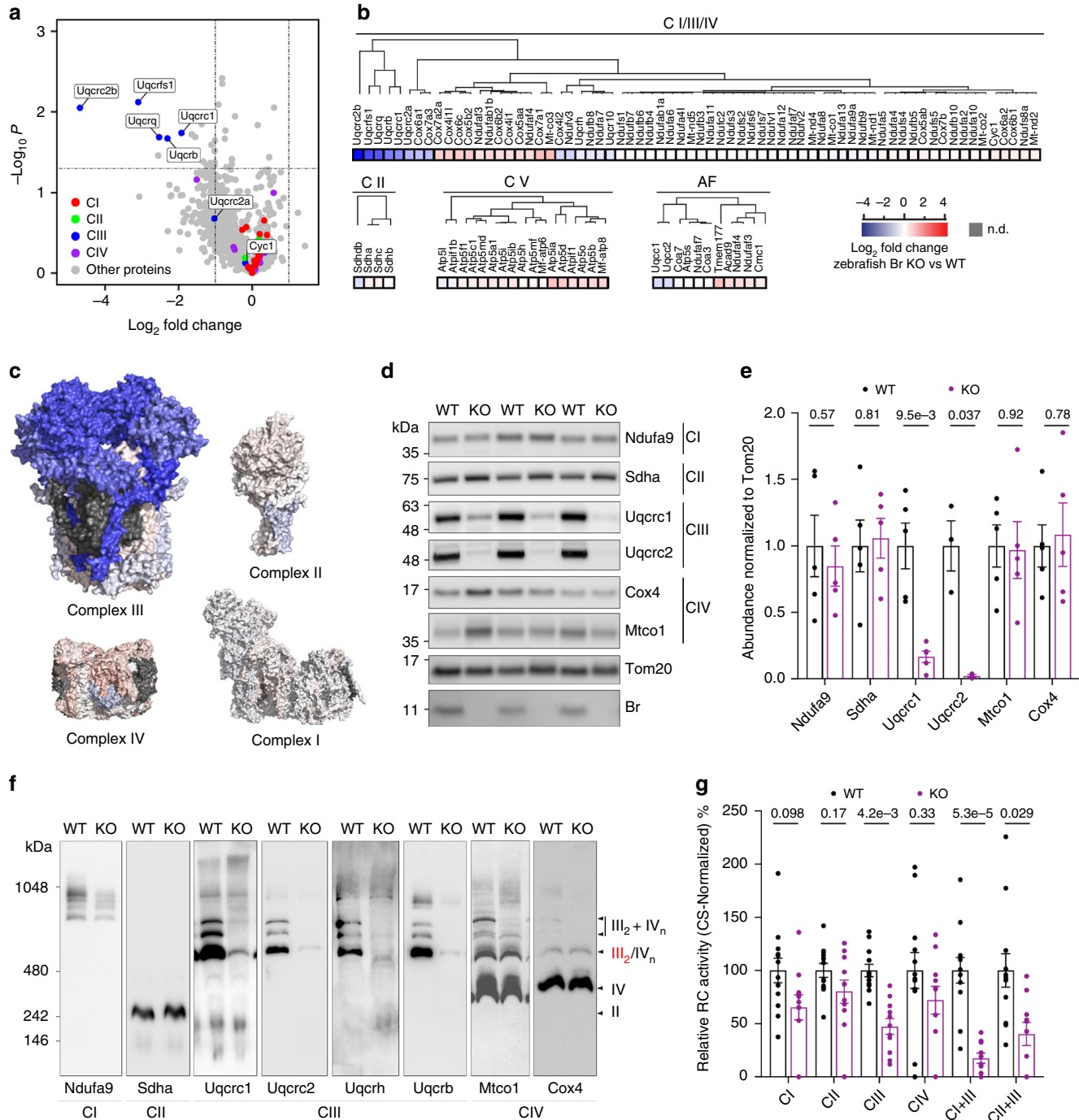

**Fig. 6 Br is required for complex III assembly and activity. a** Quantitative proteomic analysis of WT and *br* zKO isolated adult skeletal muscle (SKM) mitochondria. Mean values of log₂ fold change (zKO/WT) for each detected protein from three biological replicates are plotted in a volcano plot. Subunits of each of the ETC complexes are labeled in the indicated colors. CI–CV = Complex I–Complex IV. *p*-values from two-sided unpaired *t*-test. **b** The median log₂ fold change (zKO/WT) of all detected ETC proteins are further depicted in a dendrogram heatmap and separated by complex membership. AF = assembly factors. **c** The median log₂ fold change (zKO/WT) of CI–CIV proteins were superimposed onto the structures of human CI–CIV to visually depict the observed changes. The same scalebar from panel 6b applies. Undetected proteins are shown in gray. **d** SDS-PAGE of purified WT and zKO SKM mitochondria followed by immunoblotting for the indicated ETC proteins. Image is representative of two experiments. **e** Quantitation of SDS-PAGE in 6d. Data represent mean and SEM. *n* = 5, except for Uqcrc2 where *n* = 3. *p*-values from two-sided unpaired *t*-test. **f** BN-PAGE and western blot analysis of WT and zKO SKM mitochondria. Position of free III₂ (red) and III₂ + IVₙ (black) supercomplexes are indicated by arrowheads. **g** Spectrophotometric respiratory chain activity assay (RCA) of WT and zKO SKM mitochondria. Data represent mean and SEM. Each dot represents one animal, *n* = 12 for WT and zKO animals. *p*-values from two-sided unpaired *t*-test.

BR levels affect cellular AMPK status. This raises the possibility that BR, through regulation of AMPK activation, might also be involved in regulating intra-hepatic lipid accumulation[40], a hypothesis that remains to be investigated. Recently, CIII integrity has been implicated in regulatory T cell function and the prevention of autoimmunity[41], raising the intriguing possibility that BR can also coordinate nutrient availability with immune function. Lastly, congenital CIII defects resulting from mutations in CIII subunits or assembly factors cause very rare but severe mitochondrial disease characterized by growth retardation, lactic acidosis, aminoaciduria and early death[42,43]. The phenotypic similarities of the *br* KO zebrafish sets the basis to screen unsolved cases of mitochondrial disease for pathogenic variants in the *BR* (*C12orf73*) gene and potentially the genes encoding the other mito-SEPs described here.

In conclusion, by performing a screen of the SEP peptidome, we report the discovery of BRAWNIN, a mitochondrial peptide that is central to vertebrate oxidative phosphorylation. This emerging theme of the preponderance of SEPs in the mito-chondria is seen also in the human heart[8], pointing to the in vivo significance of mito-SEPs in human physiology. When this manuscript was in revision, Saghatelian and colleagues reported the independent discovery of PIGBOS (MSUF1 in this study) as a mitochondrial peptide that mediates the unfolded protein response in the endoplasmic reticulum[44], underscoring the bio-logical diversity of mito-SEPs. Of high priority is deducing the contribution of the other 14 mito-SEPs to oxidative metabolism. Our work demonstrates the promise and value of mining and characterizing the sORF-encoded peptidome, which should motivate similar efforts to functionally characterize remaining uncharacterized SEPs, also known as Small GEms hidden in Plain view.

## Methods
**sORF selection criteria**. sORF-encoded peptides (SEPs) were filtered using ribo-some profiling-based metrics derived from sORFs.org, adapting the guidelines from Olexiouk et al.[45]. In all, 2343 human SEPs encoded by annotated short CDS genes, lincRNAs, upstream ORFs (uORFs), and intergenic ORFs that passed ribosome sequencing quality thresholds were selected for mitochondrial prediction (Ensembl annotation bundle v92 is applied). From the pool of 115,223 predicted sORFs, a minimum ORF-score of 6[46] was imposed on nucleus encoded sORFs. Extra fil-tering criteria are set separately dependent on the sORF annotation. For the sORFs annotated in CDS genes, a minimum threshold of 50% in-frame coverage and a coverage uniformity between +1 and −1 was imposed, whereas for sORFs located on lincRNA, an in-frame coverage of 50% and a coverage uniformity between −0.5 and 0.5 was imposed. For sORFs located in the 5′-UTR region of protein coding genes, a minimum in-frame coverage of 0.7, a coverage uniformity between −0.3 and 0.3, a minimum length of 20 a.a. and a good FLOSS categorization was imposed[2]. Also, these sORFs should be identified/predicted in at least 10 different RIBO-seq datasets. For intergenic sORFs (i.e. in between genes), a minimal up- and down-stream gene distance of 1000 nucleotides (>1 kb_nn), a minimal in-frame coverage of 0.9, a coverage uniformity between −0.3 and 0.3, a minimum length of 20 a.a., a good FLOSS categorization, and a minimum number of identifications in at least 1 RIBOseq dataset was imposed. This resulted in a pool of 2343 sORFs encoded from 1027 different genes (Fig. 1C). See Supplemetary text for more explanation of selection parameters.

**Mitochondrial prediction by WGCNA and GSEA**. RNA-seq datasets from three human tissues known to be enriched in mitochondrial activity were selected to enable mitochondrial functional prediction: non-neoplastic liver (21 samples from GSE94660 – RPKM normalized), normal heart left ventricular tissue (92 samples - VST counts)[47] and normal skeletal muscle (84 samples from GSE120862 – DESeq normalized counts). To compare those mitochondria-enriched tissues with a non-enriched one, a sun-protected skin dataset (91 samples from GSE85861 – Normalized counts) was selected. To clean the datasets, genes with at least one read in more than half of the samples were kept except for the derived dataset. The healthy heart dataset is coupled to a dilated cardiomyopathy (DCM) dataset and genes with at least one read in more than 5% of the samples were kept. To distinguish between a mitochondrial and a non-mitochondrial gene, we selected 1158 bona fide mitochondrial genes ("mito") (from Human MitoCarta2.0) for comparison against 957 nuclear transcription factors (TF) randomly selected from www.tfcheckpoint.org ("non-mito")[16,48]. These form the "classifier genes". We further selected 56 known MitoCarta genes that encoded

proteins <100 a.a. (i.e. mito-SEPs) and 83 annotated nuclear and secreted proteins <100 a.a. (i.e. non-mito SEPs) which formed the mito vs non-mito SEP. These form the "training genes". We performed WCGNA with the WGCNA package in R[26,49] for all classifier and training set genes in all three datasets. i.e. we ranked all genes in a dataset according to their Spearman's correlation with each test gene. Next, we per-formed GSEA[50] using 7 gene set collections from MSig DB (Broad Institute)[51,52]: c2.cp.reactome, c2.cgp, c2.cp.kegg, c5.cc, c5.bp, c5.mf, and h.all, which gather 10,260 gene sets. Using fgsea package in R[53], the NES score of each gene set was calculated for each ranked list. NES was set to 0 if the GSEA output p-value (using an empirical phenotype-based permutation test procedure[50]) was not significant ($p > 0.05$). To decrease noise and increase specificity to mitochondrial activity, only gene sets with at least 100 mito genes with an absolute NES above three were kept and gene sets containing more than 500 genes were removed. On average, each gene had 63, 85, and 118 NES values in the heart, liver and SKM dataset respectively. These scores were used as variables for principal component analysis (PCA) using stats package. Alto-gether, each PCA analysis contained 3090 observations (1956 classifier+139 training +995 SEP candidate genes). Only 995 of the 1027 genes encoding 2343 SEP candi-dates were expressed in any one of the three datasets and could be tested by this method. Next, K-means clustering was applied to the PCA for a cluster number varying from 2 to 15. For each iteration, clusters were ranked from the highest percentage of mito genes to the lowest. To know if a cluster should be considered as positive or negative for mitochondrial identity, each cluster's mito percentage was tested as a potential threshold. The threshold which led to the highest number of correct prediction of the training set was kept. Clusters with a mitochondrial per-centage below the threshold were considered negative, otherwise positive. Candidates were given a binary score of 0 or 1 depending on whether they belonged to a negative or positive cluster respectively. This process of binary scoring was made for every cluster number, and each SEP candidate was given an average k-means score. K-mean score was set to −1 if the gene was not expressed in the dataset. Based on the k-means scores of the training set, it appeared that 75% was the minimum average k-means score needed for a peptide to be predicted as mitochondrial. To be more stringent on the selection we added the criteria of confidence level. "High confidence" meant that a SEP candidate required an average k-means score above 75% in all RNAseq where it is expressed. Refer to Supplemental Text for more information.

**Mitochondrial prediction by targeting motif prediction**. Mitochondrial targeting motifs were predicted in peptides that were not selected by WGCNA/GSEA. Dif-ferent types of mitochondrial targeting motifs were predicted using separate algorithms. Classical mitochondrial targeting sequences (MTSs) were captured by the consensus prediction of TargetP (mitochondria, Reliability class 1 & 2) and Mitofate (score > 0.8) or Mitoprot (score > 0.8)[54–56]. Transmembrane domains (TMDs) with modest hydrophobicity also serve as mitochondrial targeting signal, but they are poorly predicted by algorithms used above because these sequences are not cleavable. These hydrophobic α-helices were predicted by TMHMM or SignalP (4.1 and 5)[54,57]. To eliminate secretory proteins, the maximum TMD hydro-phobicity was set at 3[58], as we noticed that mitochondrial TMDs were on average less hydrophobic compared to non-mitochondrial TMDs[59]. In addition, mito-chondria intermembrane proteins often contain twin cysteine pairs (two C9XC, two C3XC, or one C9XC and one C10XC, where X refers to any residues in between two cysteine residues)[60], and these internal signals are generally missed by prediction algorithms. We thus included peptides with these characteristic cysteine pairs in the screen.

**Mitochondrial prediction by mitochondrial peptide RESPIN**. Quantitative mass spectrometry data on isolated mitochondria are available at the PRIDE public proteomics data repository (accession number PXD004666)[61] and downloaded using the PRIDE API[62,63]. Subsequently, the raw data files were converted and filtered using MSconvert[64]. Next, the SearchGui tool[65] was used to perform the peptide to spectrum matching, based on a combination of two database search algorithms X!tandem[66] and MSGF+[67] against a custom sequence database. This database consists of (i) the UniProt human functional proteome including all isoforms[68], (ii) the human sORFs from the sORFs.org database[69], and (iii) the cRAP database (https://www.thegpm.org/crap/). Search engine parameters were adapted from the original study[61]. Matched PSMs were validated using Peptide-Shaker[70], only PSM solely mapping to sORF peptide sequences with a spectrum coverage of at least 30% and FDR<0.01 were retained. After selecting peptides by the three strategies, redundant isoforms were manually removed if they share identical sequences with the longest isoform encoded by the same gene, with ribosomal periodicity information been considered.

**SEP transient over-expression**. The open reading frames (ORFs) of selected can-didates were synthesized and inserted into the NheI-XhoI site of a pcDNA3.1 (+) vector, under the control of the CMV promoter. Peptides were tagged with a single HA tag distal to their predicted targeting motifs or c-terminus if no motif was predicted. ORFs with non-AUG start codons were replaced with a AUG start codon to avoid the need for including cognate 3′UTRs required for recognizing near-cognate start sites[71]. HeLa cells were seeded in 24-well plates and transfected with FuGENE (Promega) and stained as described in the immunofluorescence section.

**Primer sequences**. List of primers used in this study:

| Name | Purpose | Sequence (5′ to 3′) |
| --- | --- | --- |
| LHW4_zfBr_5′_F | Br Zebrafish genotyping | AAAATTTCTCATCCTTAGCATAACATC |
| LHW7_zfBr_3′_R | Br Zebrafish genotyping | AGAACTTGCAGTTGGCATCA |
| LHW384_msBr_P1 | Br Mouse genotyping | TCAGGCCATCTGAAGGCTAC |
| LHW385_msBr_P2 | Br Mouse genotyping | TCCCGCGATACAGTTTCTCT |
| LHW386_msBr_P4 | Br Mouse genotyping | CCATTGAGCCCCCTAAATTC |

**Zebrafish Br knockout generation**. zfBrawnin is encoded by the si:ch211-68a17 gene which contains 3 exons with the ORF in exons 2 and 3. EST mining on NCBI and ENSEMBL mapping to this region in the zebrafish genome did not reveal the presence of a duplicated allele or paralogue. To completely excise the ORF, we used ZiFiT[72] to design two gRNAs 5′ and 3′ to the ORF with recognition sites underlined. ATG in bold marks the translation start site.

5′ zfBR_gRNA:

5′-GAAATTAATACGACTCACTATAGG**ATG**CCAGCAGGCGTATCTGTTT TAGAGCTAGAAATAGCGTTTTAGAGCTAGAAATAGCAAGTTAAAATAAG GCTAGTCCGTTATCAACTTGAAAAAGTGGCACCGAGTCGGTGCTTTT -3′

3′-zfBR_gRNA:

5′-GAAATTAATACGACTCACTATAGAACTGCGAACAGAACTTCTGTTT TAGAGCTAGAAATAGCAAGTTAAAATAAGGCTAGTCCGTTATCAACTTG AAAAAGTGGCACCGAGTCGGTGCTTTT-3′

gRNAs were generated from gBLOCKS with MEGAshortscript[TM] T7 Kit (Invitrogen, AM1354). Following DNaseI treatment and precipitation with NH$_4$Ac and ethanol, gRNAs were resuspended in 40 μl pure water. In all, 200 pg of each gRNA was injected along with 300 pg of purified recombinant Cas9 protein (a kind gift of Harwin Sidik) into 1 cell stage AB larvae. F0 founders were detected by genotyping with the primers LHW4_zfBr_5′_F and LHW7_zfBr_3′_R.

WT undeleted band yields a 624-bp band while the deleted allele (brΔ) yields a 220-bp with HotStart Taq Polymerase (Qiagen). In all, 2 brΔ founders (ΔBig and ΔSmall) with complete deletions were brought forward for characterization. Both were backcrossed to AB until F2 before phenotypic investigations began. All data showed in the paper are generated from F2 and F5 incrosses. Growth retardation phenotype is stable throughout generations and reproducible to F6 when this manuscript was prepared. Both founders showed similar growth defects. All data presented are derived from ΔBig but are representative of ΔSmall, which had a more severe phenotype.

**Zebrafish husbandry and growth tracking**. Zebrafishes were reared under standard conditions, 28 °C with approved protocols following regulations stipulated by the IACUC committee of A*STAR Singapore. Heterozygous crosses were performed starting at the F2 generation to produce WT, Hets, and zygotic KO larvae. These were housed at a density of 50 per tank. In all, 15–20 animals were randomly selected and genotyped every week by scaling to track Mendelian representation over time under conditions of unsegregated growth. Briefly, scales were lysed in 50 mM NaOH at 95 °C for 15 min followed by neutralization with 1/ 10th volume Tris-HCl pH 8.0 and debris pelleted by centrifugation at 3000 × g for 5 min before PCR with LHW4_zfBr_5′_F and LHW7_zfBr_3′_R primers listed above. To enable segregated growth, larvae were genotyped at 3dpf by clipping the tip of the tail fin[73] and housed at a lower density of 20/tank. KO fish were given one extra feed per day to boost their growth. The standard length of these larvae were measured once every week under a stereoscope (up to day 28) or with a ruler after minimal tricaine anesthetization. Br KO fish that survive past 4 months have a short mating span of about 2 months to generate maternal zygotic (mz KO) larvae. Beyond that point, we notice that the Br KO males fail to display effective mating behavior due to their low body weight and lethargic phenotype. Female Br KO females are however, still able to mate with WT or Het fish, demonstrating preservation of their fertility.

**Birefringence analysis**. In all, 4–6 dpf larvae were anaesthetized briefly with tricaine before embedding in 30% methylcellulose in the egg water. Imaging was performed by placing the dish containing the larvae in between a pair of polarized filters (50 mm Diameter Unmounted, Linear Glass Polarizing Filter, Stock No. #43-787 from Edmund Optics) aligned at 90° to each other. Images were collected under brightfield with a Nikon SMZ800 stereoscope connected to a CCD camera.

**Zebrafish motility monitoring**. In all, 5 dpf mz KO and WT larvae were gently pipetted into a 48-well plate, each fish per well for monitoring locomotor activity. The 48-well plate was placed in a sound-attenuated incubator to isolate it from extraneous noise and light. Inside the incubator, a while-light LED box was

positioned below the plate and videos were recorded from the top at a speed of 5.3 frames per second (5.3 fps) by a Basler USB3 camera in a resolution of 2048 × 1024 pixels. The position (x-y coordinates) of the fish was determined in real-time based on background subtraction algorithms written in Python utilizing OpenCV library. This data was then exported into Microsoft Excel files and analyzed offline for speed and distance measures.

**Zebrafish respiratory analysis**. Manually dechorionated 24 hpf (1 dpf) mz KO larvae were placed into each well (1 per well) of an Agilent XF96e Spheroid microplate containing 180 ul of sea salt water and allowed to adapt for 1 h. Basal respiration was recorded on the Seahorse XF96e at 28 °C. This was achieved by placing the XF96e in a refrigerated chamber held at 16 °C. We were unfortunately unable to perform a Mito Stress test on live larvae because the repeated mixing steps following injection of Oligomycin and FCCP caused significant lethality to the larvae due to the low height clearance of the XF96e oxygen probe.

**Zebrafish metabolomic analysis**. At 5 dpf, 50 WT or mz KO larvae were collected into a tube under normal conditions, placed on ice for 5 min to immobilize larvae, followed by gentle sedimentation. All egg water was removed and larvae were rinsed washed twice with ice cold water. The dried pellet was weighed to 2 decimal points in mg and snap frozen, thawed on ice and homogenized in 50% H$_2$O/50% Acetonitrile using zirconia beads in a liquid nitrogen cooled rotor (2 cycles of 5000 × g for 20 s). For organic acid extraction, 300 μL of tissue homogenate was extracted with ethylacetate, dried and derivatized with N, O-Bis(trimethylsilyl)trifluoroacetamide, with protection of the alpha keto groups using ethoxyamine (Sigma Aldrich, USA). The mixture was allowed to equilibrate for 30 s and 1.2 mL of HPLC grade methanol was added to the mixture. After vortexing, the mixture was incubated in 50 °C for 10 min and centrifuged to pellet the precipitated protein. The supernatant was removed and dried under nitrogen gas in a clean microcentrifuge tube. The dried extract was reconstituted in 100 μl of methanol before analyzing using a liquid chromatography–mass spectrometer. Trimethylsilyl derivatives of organic acids were separated by gas chromatography on an Agilent Technologies HP 7890A and quantified by selected ion monitoring on a 5975C mass spectrometer using stable isotope dilution. The initial GC oven temperature was set at 70 °C, and ramped to 300 °C at a rate of 40 °C/min, and held for 2 min.

**MitoCore modeling**. The Mitocore model was downloaded in SBML format from Zieliński et al.[32]. Each of the 485 reactions in the model was blocked (setting upper and lower bounds of permissible flux to zero), one reaction at a time, and the ATP production (Reaction ID: "OF_ATP_MitoCore") was constrained to have a flux >90% of the theoretical capacity of the blocked model. Uniform sampling was performed for each blocked model using the CHRR algorithm of Cobra toolbox 3.0 to obtain 5000 flux vectors that represent the solution space. As baseline comparison, the unchanged MitoCore model (unconstrained model) was also sampled to obtain 5000 flux vectors. The flux distributions for the export reactions of Lactate (Reaction ID: 'L_LACt2r') and Succinate (Reaction ID: 'SUMt_MitoCore') were compared between the unconstrained model and each of the 485 constrained models using the Wilcoxon ranksum test, and the effect sizes were computed using Cohen's d. Reaction blockages that produced a negative effect size for either lactate or succinate export were discarded, and the remaining blocked models were ranked by the sum of effect sizes (for Lactate and Succinate). A similar workflow was repeated to identify which of the 371 genes in the MitoCore model had largest effect size to increase succinate and lactate export when knocked out (Single Gene Deletion studies using Cobra Toolbox 3.0)[74].

**Mitochondria flux assays**. Respirometric analysis of purified mitochondria from zebrafish skeletal muscle was performed using Agilent's Seahorse platform as described by Boutagy et al.[75]. Briefly, 1.0–1.4 μg of purified mitochondria were plated in each well of a XF96e cell culture plate. To measure uncoupled electron flow, assay buffer contained 5 mM pyruvate, 4 μM carbonyl cyanide 4-(trifluoromethoxy) phenylhydrazone (FCCP), 1 mM malate. Rotenone (2 μM, final), succinate (5 mM, final), antimycin A (4 μM, final), TMPD/ascorbic acid (100 μM/ 10 mM) were injected sequentially during the assay. In coupling assay with pyruvate and malate, assay buffer contained 10 mM pyruvate and 5 mM malate. In coupling assay with Complex I inhibited, assay buffer contained 10 mM succinate and 2 μM rotenone. In coupling assay, ADP, oligomycin A, FCCP and antimycin A were injected into assay plate sequentially to 3.2 mM, 3 μM, 4 μM and 4 μM, respectively. Citrate synthase activity (described below) was used for postnormalization of OCR rates.

**Mitochondria purification from cultured cells and tissues**. Mitochondria were isolated from cultured cells using mitochondria isolation kit (Abcam, ab110168). Cells were harvested and ruptured with a Dounce homogenizer in the buffer A containing cOmplete protein inhibitor cocktail (Roche) and 1 mM phenylmethylsulfonyl fluoride (PMSF). 15 stokes were performed with a motorized pestle operated at 300 rpm. Nuclei and unbroken cells were removed by centrifugation at 600 × g. Homogenates were spun at 7000 × g for 15 min to precipitate mitochondria. Mitochondria pellets were collected for downstream uses.

To isolate mitochondria from zebrafish, skeletal muscles of fish were dissected and minced. Tissue was disrupted in muscle mitochondria isolation buffer (67 mM sucrose, 50 mM KCl, 1mM EDTA, 0.2% fatty acid free BSA, 50 mM Tri-HCl, pH7.4) with a Dounce homogenizer tight pestle operated at 1300 rpm. The extract was centrifuged at $600 \times g$ for 5 min to clear intact myofibrils and heavy cell debris and then spun at $7000 \times g$ for 15 min to isolate the desired mitochondrial fraction.

**Antibody generation.** Custom polyclonal antibodies were generated against human BR (hBR) and zebrafish BR (zBr) using the following peptide antigens.

For SDS-PAGE, BN-PAGE, IF and IHC of human/mouse BR: 5′-EVVHRYYRPDLTIPEIPPKRGELKTELLGLKERKHKPQVSQQEELKC-3′

For SDS-PAGE of zBr: 5′-RPDLSIPEIPPKPGELRTELLGLKERQMDSQKQ-3′

For BN-PAGE of zfBr: 5′-RPDLSIPEIPPKPGELRTEC-3′

Peptides were conjugated to KLH and used for rabbit immunization according to standard procedures.

**Immunofluorescence and immunohistochemistry.** Cells were fixed by 4% PFA at 37 °C for 20 min and washed with PBS twice. Where appropriate, cells were stained with 25 nM of Mitotracker Red at 37 °C for 30 min before the fixation. Samples were solubilized with 0.3% Triton X at room temperature for 10 min and washed twice with PBS. Cells were blocked with blocking buffer (3% BSA, 10% FBS in PBS) at room temperature for 1 h. Primary antibodies were diluted to desired concentrations in the blocking buffer and incubated with cells for over-night, rocking at 4 °C. Proteins of interest were probed with the following antibodies: anti-BR, 1:200 (Novus, NBP1-90536); anti-HA, 1:500 (BioLegend, 901502); anti-CYTC, 1:500 (Santa Cruz, sc-13561); anti-CS, 1:500 (Santa Cruz, SC-390693); anti-TOMM20, 1:500 (Proteintech, 11802-1-AP). Cells were washed with three changes of PBS-Tween 20 0.1% (v/v) before incubating with 4 μg/mL of secondary antibodies for 1 h at room temperature (Alexa fluro 488, 594, 647, Invitrogen). In mouse skeletal muscle staining, cryosections were air dried and fixed with 4% formaldehyde in PBS at 4 °C for 10 min Samples were solubilized and blocked in PBS containing 0.3% TX-100 and 5% normal goat serum at 4 °C for 1h. Primary antibodies were prepared in PBS containing 1% normal goat serum and 0.1% TX-100 and incubated for over-night at 4 °C. anti-BR, 1:200; anti-CS, 1:500 (Santa Cruz, SC-390693) were used for muscle staining. Slides were washed times with PBS containing 0.1% TX-100 and incubated with 4 μg/mL of secondary antibodies for 1 h at room temperature (Alexa fluro 488, 594, 647, Invitrogen). Samples were imaged on an Olympus FV3000 confocal laser scanning microscope. FFPE sections of deidentified human cardiac and muscle tissues were purchased from US Biomax, dewaxed and stained according to standard procedures with anti-BR 1:200 followed by anti-rabbit HRP.

**Immunoblotting.** Whole cell or mitochondrial samples were lysed in RIPA buffer supplemented with cOmplete protein inhibitor cocktail (Roche) and 1 mM phenylmethylsulfonyl fluorid (PMSF). When appropriate, protein concentration was measured by bicinchoninic acid (BCA) assay (Thermo Fisher). Lysates were boiled in 1X laemmli sample buffer (50 mM Tris-HCl pH6.8, 2% SDS, 10% glycerol, 12.5 mM EDTA, 0.02% bromophenol blue, 50 mM DTT). After resolving by SDS-PAGE in NuPAGE MES SDS running buffer (Thermo Fisher), proteins were transferred to a 0.2-μM PVDF membrane (Thermo Fisher) and incubated in blocking buffer (3–5% non-fat milk TBS-Tween 20 0.1% (v/v)) for 1 h at room temperature. Membranes were probed with appropriate antibodies in TBST milk for overnight, rolling at 4 °C. Following antibodies were used: anti-BR, 1:3000 (Novus, NBP1-90536); custom anti-human BR (1 μg/mL); custom anti-zebrafish BR (1 μg/mL); anti-ACTIN, 1:5000 (Sigma, A2228); anti-TOMM20, 1:5000 (Proteintech, 11802-1-AP); anti-CS, 1:2000 (Santa Cruz, SC-390693); anti-ATP5A, 1:5000 (Santa Cruz, SC-136178); anti-TOMM70, 1:5000 (AbClonal, A4349); anti-VDAC1, 1:2000 (AbClonal, A0810); anti-MTCOI, 1:3000 (Abcam, ab14705); anti-TIM23, 1:5000 (Proteintech, 11123-1-AP); anti-OXPHOS cocktail, 1:5000 (Invitrogen, 45-7999); anti-UQCRC1, 1:5000 (AbClonal, A3339); anti-NDUFA9, 1:3000 (AbClonal, A3196); anti-SDHA, 1:5000 (AbClonal, A2594); anti-TUBULIN, 1:5000 (Sigma, T9822); anti-ACC, 1:2000 (CST, 3662S); anti-pACC, 1:2000 (CST, 3661S); anti-GAPDH, 1:5000 (CST, 2118S); anti-FLAG, 1:2000 (BioLegend, 637302); anti-C4ORF48 (Abcam, ab185315); anti-COX4, 1:1000 (AbClonal, A10098); anti-UQCRC2, 1:5000 (Abclonal, A4181); anti-UQCRFS1, 1:2500 (Abcam, Ab14746); anti-UQCRH, 1:2500 (Abcam, ab134949); anti-UQCRB, 1:2000 (Sigma, HPA043060); Anti-MTCYB, 1:2000 (Proteintech, 55090-1-AP); Anti-AMPK, 1:3000 (Bio-Rad, VMA00246); and Anti-pAMPK, 1:3000 (CST, 2535). Membranes were extensively washed with four exchanges of fresh TBST and probed with HRP conjugated secondary antibodies against mouse, rabbit or rat IgG at 1:5000 (Jackson ImmunoResearch) and washed again. Chemiluminescence was captured by X-ray films (Santa-Cruz) or a Chemidoc imaging system (Bio-Rad). For fluorescence blots, similar procedures were followed except that primary antibodies and secondary antibodies was prepared in 3% BSA. IRDye 800CW goat anti-rabbit and IRDye 680RD goat anti-mouse (Licor) were used at 1:10,000. Fluorescence blots were scanned using Licor Image Studio.

**Blue-native PAGE analysis.** BN-PAGE analysis was performed as described by Jha et al.[76] and 2nd dimension BN-SDS PAGE as described by Fiala et al.[77] with minimal modifications. Isolated mitochondria were solubilized in BN-PAGE sample buffer (Thermo fisher) with 8 g/g of digitonin on ice for 20 min. Every 50 μg of mitochondrial proteins were solubilized in 20 μl of buffer. The insoluble fraction was removed by centrifugation at $20,000 \times g$ for 20 min 20 μl of solubilized protein lysate was stained with 2.5 μl of 5% G-250 sample additive (Thermo fisher). Wash wells of Native-PAGE 3–12% Bis-Tris protein gels with the dark blue cathode buffer and load 50 μg of mitochondrial proteins into each lane. Fill the inner chamber with the blue cathode buffer (Thermo fisher) and the outer chamber with the native running buffer (Thermo fisher). Protein complexes were separated by electrophoresis for 30 min at 150 V. After changing the buffer in the inner chamber to light blue buffer, run for another 60 min at 250 V. Resolved protein complexes were transferred to a PVDF membrane using the high molecular protein default program on a Trans-Blot Turbo Transfer System (Bio-Rad). The membrane was fixed in 8% acetic acid and destained in methanol. The membrane was blocked and probed for proteins of interest as described in the immunoblotting section.

In the 2nd dimension BN-SDS-PAGE analysis, gel stripes of BN-PAGE gel were excised and incubated in 1X laemmli sample buffer and incubated at 95 °C briefly. The gel stripe was horizontally inserted into a SDS-PAGE gel in which gel fingers were removed. The rightmost gel finger was kept for the purpose of loading a SDS treated control sample. Proteins were resolved by SDS-PAGE and analyzed by immunoblotting as described above.

**Sodium carbonate extraction.** Sodium carbonate extraction was performed as described with minor modifications[78]. Mitochondria pellets isolated from HEK293T cells were resuspended in buffer C (320mM Sucrose, 1 mM EDTA, 10 mM Tris-Cl pH 7.4), buffer C containing 1% Triton X or 0.1M $Na_2CO_3$ pH 11. The resuspensions were incubated at 4 °C for 1 h with gentle nutation. Supernatant and insoluble pellet were separated by centrifuge at $20,000 \times g$ for 30 min, and the supernatant was gently removed without disturbing the pellet. Equal fractions of supernatant and pellet were analyzed by SDS-PAGE immunoblotting.

**Mitochondrial membrane fractionation.** Mitochondria were isolated with iso-tonic buffer (250 mM sucrose, 1 mM EDTA, 10 mM HEPES-KOH pH 7.4, protease inhibitor cocktail and 1 mM PMSF) from HEK293T cells following procedures described in the mitochondria isolation section. Mitochondria pellet was gently resuspended in the isotonic buffer to get protein concentration at 1 mg/mL. Aliquots of mitochondria were solubilized by increasing concentrations of digitonin (0%, 0.1%, 0.115%, 0.13%, 0.145%, 0.16%, 0.175%, 0.19%, 0.205%, and 0.22%) for 1 h at 4 °C. Soluble and insoluble fractions were separated by centrifugation at $20,000 \times g$ for 20 min Insoluble fractions were fully resuspended in an equal amount of isotonic buffer to match the volume of supernatant. Both fractions were lysed in 1X laemmli sample buffer and analyzed by SDS-PAGE and immunoblotting.

**Protease sensitivity assay.** Mitochondria were isolated with isotonic buffer (250 mM sucrose, 1mM EDTA, 10 mM HEPES-KOH pH7.4, protease inhibitor cocktail) from HEK293T cells. Mitochondria pellet was washed to eliminate protease inhibitors. Mitochondria were resuspended to a protein concentration of 1 mg/mL. Mitochondrial membranes were by differentially solubilized by increasing concentrations of digitonin (0–0.2% as indicated in the figure) or 1% Triton X for 10 min on ice. Proteinase K was then added to 100 μg/mL and incubated for 30 min on ice to allow the complete digestion of accessible proteins. To terminate the protease digestion, PMSF was freshly prepared and added to a concentration of 8 mM. Samples were analyzed by western.

**Mass spectrometry analysis of isolated mitochondria.** Crude mitochondrial protein isolated from adult zebrafish skeletal muscle or MEF was normalized using the Pierce Protein Assay Kit (ThermoFisher Scientific). Mitochondrial protein was solubilized in 1% (w/v) SDC, 100mM Tris pH 8.1, 40 mM chloroacetamide (Sigma) and 10 mM tris(2-carboxyethyl)phosphine hydrochloride (TCEP; BondBreaker, ThermoFisher Scientific) for 5 min at 99 °C with 1500 rpm shaking followed by 15 min sonication in a water bath sonicator. Protein digestion was performed using trypsin (ThermoFisher Scientific) at a 1:50 trypsin:protein ratio at 37 °C overnight. The supernatant was transferred to Stagetips made in-house containing 3x14G plugs of 3M™Empore™ SDB-RPS substrate (Sigma) as described previously[61,79] with modifications. Briefly, isopropanol (99% v/v) containing 1% (v/v) TFA was added to the tip and mixed with the lysate prior to centrifugation at $3000 \times g$. Stagetips were then washed first with the same solution followed by an additional wash with 0.2% (v/v) TFA. Peptides were eluted in 80 % (v/v) acetonitrile (ACN) and 1% (w/v) NH4OH, then acidified to a final concentration of 1% TFA prior to drying in a CentriVap Benchtop Vacuum Concentrator (Labconco). Peptides reconstituted in 0.1% TFA were desalted on Pierce C-18 tips (Thermo Scientific) as per manufacturer's instructions prior to drying under vacuum. Desalted peptides were reconstituted in 100mM triethylammonium bicarbonate (TEAB, Thermo-Fisher Scientific) and labeled with 6plex Tandem Mass Tags (TMT) (ThermoFisher Scientific) in 8:1 label:protein ratio as per manufacturer instructions. Labels were pooled such that each multiplex set contained three replicates each of Br KO and

control mitochondria. Pooled samples were fractionated using the Pierce High pH Reversed-Phase Peptide Fractionation Kit (ThermoFisher Scientific) as per manufacturer's instructions for a TMT experiment. Individual fractions were dried under vacuum and reconstituted in 2% (v/v) acetonitrile (ACN) and 0.1% (v/v) trifluoroacetic acid (TFA).

Liquid chromatography (LC) coupled MS/MS was carried out on an Orbitrap Lumos mass spectrometer (ThermoFisher Scientific) with a nanoESI interface coupled to an Ultimate 3000 RSLC nanoHPLC (Dionex Ultimate 3000). The LC system was equipped with an Acclaim Pepmap nano-trap column (Dionex-C18, 100, 75 μm × 2 cm) and an Acclaim Pepmap RSLC analytical column (Dionex-C18, 100., 75 μm × 50 cm). The tryptic peptides were injected to the trap column at an isocratic flow of 5 μL/min of 2% (v/v) ACN containing 0.1% (v/v) formic acid for 5 min applied before the trap column was switched in-line with the analytical column. Peptides were eluted using 5% DMSO in 0.1% v/v formic acid (solvent A) and 5% DMSO in 100% v/v ACN and 0.1% v/v formic acid (solvent B) using a custom 128-minute non-linear gradient that included a step for equilibration of the column prior to injection of the next sample. The Synchronous Precursor Selection (SPS)–MS3 based TMT method was used. In brief, a full MS1 spectra was acquired in positive mode at 120,000 resolution scanning from 380–1500 $m/z$. AGC target was at 4e5 and maximum injection time of 50 ms. Precursors for MS2/MS3 analysis were selected based on a Top 3 second method. MS2 analysis consists of collision induced dissociation (CID) with isolation window of 0.7 in the quadrupole. CID was carried out with normalized collision energy of 35 and activation time of 10 ms with detection in the ion trap. Following acquisition of each MS2 spectrum, multiple MS2 fragment ions were captured in the MS3 precursor population using isolation waveforms frequency notches. The MS3 precursors were fragmented by high-energy collision-induced dissociation (HCD) and analyzed using the Orbitrap with scan range of 100–500 m/z, 60,000 resolution, normalized collision energy of 60%, AGC target of 1e5 and maximum injection time of 120 ms.

Raw files were processed using the MaxQuant platform (version 1.6.10.43)[80] and searched against UniProt zebrafish reviewed and unreviewed or mouse reviewed database containing canonical and isoform sequences (November 2019) using default settings for a TMT 6plex experiment. The proteinGroups.txt output was processed in Perseus (version 1.6.10.43)[81]. Briefly, Log2-transformed TMT reporter intensity corrected values were grouped into control and KO groups consisting of three replicates each. Values were normalized by subtraction of the median value in each column and identifications filtered to 100% valid values across all six samples and quantifications being made from more than 2 unique peptides. Zebrafish Uniprot identifiers were first matched to human orthologs using a database generated from the Zebrafish Information Network (ZFIN) database[82]. Zebrafish and mouse mitochondrial proteins were annotated based on the presence of their human ortholog or mouse gene symbol being present in either of the Mitocarta2.0[16] or Integrated Mitochondrial Protein Index (IMPI)[83] databases. The datasets were re-normalized for variance introduced by mitochondrial isolation through subtraction of the mean value of proteins with mitochondrial proteins using the subtract row cluster function in Perseus.

For volcano plots, a two-sided $t$-test was performed between KO and control groups with significance determined by Student's $t$-test ($p$-value<0.05, unpaired $t$-test). For clustering and mapping of data to Cryo-EM structures, median values for each experimental group were determined along with the intragroup standard deviation. Thresholds were determined for filtering highly variant medians based on the distrubtion of standard deviations (for the zebrafish dataset values <2 stdev were considered significant, for the mouse dataset only values <0.6 were retained). Ratios were determined for each protein (KO/control) and used for hierarchical clustering and structure mapping. For hierarchical clustering, protein ratios were subsetted based on their human orthologs presence in the indicated RC complexes. Euclidian distances (average linkage) were used for k-means clustering of rows and columns. For structural mapping, protein ratios for the same RC subunits were mapped onto the relevant chains of following PDB structures using a custom python script described previously[61]; Complex I, 5LDW[84]; Complex II, 1ZOY;[85] Complex III, 1BGY[86]; and Complex IV, 5B1A[87]. In the case of multiple isoforms present for some RC subunits in the zebrafish dataset, the most altered protein was used for the structure mapping.

**BR interactome analysis**. To determine physical interaction partners of BRAWNIN (BR), we purified BR (non-tagged and FLAG tag) with different antibodies at different expression levels (endogenous and over-expressed). To purify endogenous BR, mitochondria were isolated from HEK293T cells as described. Mitochondria were solubilized in 1 mL of IP buffer (150 mM NaCl, 1 mM EDTA, 50 mM HEPES-KOH pH 7.4) containing 1% n-Dodecyl β-D-maltoside (DDM), cOmplete, EDTA-free protease inhibitor cocktail (Roche) and 1 mM PMSF at 4 °C for 30 min with constant rocking. Homogenate was spun at 20,000 × g for 10 min at 4 °C to remove insoluble content and the supernatant was collected and incubated with BR antibody (Novus, NBP1-90536) or Rabbit IgG control for overnight. In all, 30 μl of protein A/G beads were washed with IP buffer containing 0.1% DDM for two times and incubated with the protein lysate for 2 h at 4 °C to allow antibody/protein complex binding. Non-interacting proteins were removed by four rounds of spin and IP buffer washes in the presence of 0.5% DDM. The beads were then washed with IP buffer only. Bound proteins were eluted with 0.1% Rapigest at room temperature. For the isolation of over-expressed BR, a similar workflow was

performed except that BR was transiently over-expressed in HEK293T before the sample preparation. For isolating FLAG tagged BR, we fused a single FLAG tag to the c-terminus of BR. BR and BR-FLAG were transiently overexpressed in HEK293T. Mitochondria were isolated from the two samples separately and solubilized with IP buffer (150 mM NaCl, 1 mM EDTA, 50 mM HEPES-KOH pH 7.4) containing 1% n-Dodecyl β-D-maltoside (DDM), cOmplete, EDTA-free protease inhibitor cocktail (Roche) and 1 mM PMSF. After clearing the insoluble fraction by spin, protein lysates were incubated with anti-FLAG M2 resin (Sigma) for overnight with nutation. The resin was washed as described above and eluted with 0.1% Rapigest (Waters).

To prepare for mass spectrometry, samples were treated with 100 mM triethylammonium bicarbonate (TEAB) pH 8.5 and 20 mM tris(2-carboxyethyl) phosphine hydrochloride TCEP for 20 min at 55 °C with shaking. Proteins were then alkylated by 55 mM chloroacetamide (CAA) at room temperature for 30 min in dark. In total, 3 volumes of 100 mM TEAB were added to dilute out Rapigest to allow following protease digestion. The sample was first treated by 2.5 μg of LysC (Wako) for 3 h at 37 °C with shaking and then treated with 2.5 μg of Trypsin (Promega) for over-night at 37 °C with shaking. Rapigest was hydrolyzed by adding TFA to 1% and removed by spin at 20, 000 g for 10 min The sample was desalted on a C18 cartridge (Oasis) and eluted with 50% ACN and 100 mM TEAB pH8.5, vacuum concentrated and resuspend in 100 mM TEAB pH 8.5. Peptides were labeled with 10 plex tandem mass tags (TMT) for over-night and then quenched by 0.5 M Tris-HCl pH7.4, pooled, desalted again and analyzed by an Orbitrap LC/MS[88].

**Mouse housing conditions**. In this study, C57BL6/J mice were housed on a 12-h light/dark cycle with water and normal chow diet provided ad libitum. Room temperature was maintained at 21–24 °C and relative humidity at 40–60%. All animal procedures were approved and conducted in accordance with the Duke-NUS, SingHealth Institutional Animal Care and Use Committee.

**BR co-IP with complex III**. To express FLAG tagged BR from mouse tissue, BR Adeno-associated Virus (AAV) expression construct was generated by inserting the murine BR ORF into a pscAAV2 backbone with a single FLAG peptide fused to the c-terminus of BR. AAV9-pseudotyped virus was produced by Vector Core, Genome Institute of Singapore, Singapore. In all, 10⁹ viral genomes were constituted in 10 μL PBS and administrated to the gastrocnemius muscle of 1-month-old female animals. Skeletal muscles were isolated 1–2 months after the AAV administration. Mitochondria were isolated from tissues as described above and solubilized with 1% digitonin (digitonin: mitochondria protein ratio was normalized to 4 g/g.) in IP buffer (150 mM NaCl, 1 mM EDTA, 50 mM HEPES-KOH pH 7.4) containing cOmplete, EDTA-free protease inhibitor cocktail (Roche) and 1 mM PMSF at 4 °C for 30 min with constant rocking. The insoluble fraction was removed by centrifugation at 20,000 × g for 15 min. The supernatant was incubated with 30 μl of anti-FLAG M2 resin and nutated for over-night in a cold room. The resin was extensively washed with IP buffer supplemented with 0.1% digitonin for four times. BR-FLAG and its interaction partners were eluted with 0.5 mg/mL 3XFLAG peptide (Sigma) in IP buffer at 4 °C for 30 min IP input and elute were analyzed by SDS-PAGE. In the reciprocal IP, mitochondria were isolated from HEK293T cells transiently expressing UQCRQ-FLAG and BR-HA. Protein IP was described as above.

**Cell culture**. All cell lines were obtained from American Type Cell Collection (ATCC; www.atcc.org) and cultured using standard tissue culture techniques. HEK293T (ATCC® CRL-3216™) cells were cultured in Dulbecco's High Glucose Modified Eagles Medium/High Glucose (HyClone). U87MG (ATCC® HTB-14™) cells were cultured in Minimal Essential Medium with L-Glutamine (Gibco). All media were supplemented with 10% South American sourced Fetal Bovine Serum (HyClone) and 1% of 10,000 U/mL of Penicillin-Streptomycin (Gibco). Starvation regimes were performed as follows: HEK293T cells were plated in normal DMEM (4500 mg/L glucose) overnight, followed by a complete media change to that containing 3 μM Etomoxir in complete media, media lacking FBS (Serum-) and or DMEM without glucose (GLUCOSE-).

**MEF generation**. Br KO were generated by Crispr/Cas9-mediated targeting of the mouse 1190007I07Rik gene followed by MEF derivation from F1 embryos. Briefly, a pair of guide RNAs (gRNAs : 5′-cctcctggccatgtgcgccg-3′, 5′-gaactcaaaacggagcttt-3′) along with recombinant Cas9 protein were introduced by pronuclear injection into C57BL6 fertilized oocytes. Successful excision of exons 2 and 3 containing the coding region was verified by genomic PCR. Heterozygous F1 mice were inter-crossed and embryos were harvested at E13.5 after $CO_2$ euthanasia. Uterine horns were dissected out and rinsed with cold DPBS (Hyclone) with 2% of 10,000 U/mL of Penicillin-Streptomycin (Gibco). Upon releasing, head and viscera of embryos were lysed for genotyping and discarded respectively. Remaining tissues were finely minced with blades and digested with 0.25% Trypsin-EDTA (Gibco) at 37 °C for 30 min with gentle pipetting every 10 min Trypsin was then neutralized with an equal amount of complete medium [Dulbecco's High Glucose Modified Eagles Medium/High Glucose (HyClone) supplemented with 10% FBS and 1% of 10,000 U/mL of Penicillin-Streptomycin (Gibco)]. Digested tissues were centrifuged at 1200 × g for 5 min at room temperature. Cell and tissue pellet resuspended and

passed through a 70-µm cell strainer (BD). Cells were seeded to a 10-cm dish pre-coated with gelatine and was cultured with complete medium. In sub-culturing, MEFs were detached with 0.05% Trypsin-EDTA (Gibco).

**BR over-expression by lentivirus transduction**. The human BR ORF was inserted into a pCDH lentivirus vector under the control of a CMV promoter. Plasmid was then packaged into viral particles using the pPACKH1 lentivector Packaging Kit (System Bioscience) in HEK293T cells. In all, 25% of total crude virus extracted was used to infect U87MG for 2 days before G418 (Santa Cruz) selection at 1.2 mg/mL.

**Brawnin knockdown**. siRNA-mediated knockdown of *BR* was performed in U87MG with ON-TARGET plus Human C12orf73 (728568) siRNAs (Dharmacon) with the following sequences: 5′-CAGUGGAUGUUUAGCCGAU-3′ (siBR_14) and 5′-CGACCGGACCUGACAAUAC-3′ (siBR_16). An equimolar ratio of both targeting siRNAs are mixed to achieve maximal knockdown. Control siRNA sequence is 5′-UGGUUUACAUGUCGACUAA-3′. Briefly, siRNAs are transfected at 25 nM final concentration using Dharmafect using a suspension transfection protocol for U87MG.

shRNA-mediated knockdown or *BR* was performed using the following sequences: 5′-AAAAGTTCGCGGTACAGTCTCTAGT-3′ (shBR_1) and 5′-AAAAGTGGATGTTTAGCCGATACGT-3′ (shBR_3). shRNA hairpins were cloned into a lentiviral backbone to allow stable expression. Viral particles were produced in LentiX293 and purified according to standard procedures. 500,000 U87MG cells were collected and resuspended with respective medium with 32 µg/ mL of polybrene, and added to polybrene-treated cells at a target MOI of 3. Cell-virus suspension solution was centrifuged at $1000 \times g$ for 1 h at room temperature to maximize the collision between cells and viruses. The mixture was then diluted four times with respective medium and seeded to one 6-well plate (Costar). Following recovery from viral transduction, cells were selected with 1 µg/mL of puromycin until a non-transduced control showed complete death. U87MG Scr, shBR cells were used within 10 passages of initial transduction to minimize phenotypic drift due to compensation.

**Seahorse cultured cell MitoStress test**. In all, 15,000 U87MG cells were plated on poly-L-Lysine (Sigma) treated Seahorse XF96 Cell Culture Microplates (Agilent) and XFe96 sensor cartridge (Agilent) was hydrated with distilled water at 37 °C with no $CO_2$ the day prior to MitoStress Assay. On the day of actual MitoStress Assay, cultured cells were washed once and media were replaced with XF basal DMEM supplemented with 1 mM pyruvate (Sigma), 2 mM glutamine (Sigma), and 10 mM glucose (Sigma). Pre-hydrated sensor cartridge was calibrated with Sea-horse XF Calibrant (Agilent). Both cell culture plate and sensor cartridge were incubated at 37 °C with no $CO_2$ for at least 45 min before actual assay starts. For MitoStress assay, 2 µM of oligomycin (Sigma), 1 µM of FCCP (Sigma), and 1 µM of Rotenone/Antimycin (Sigma/Sigma) were injected according to Seahorse MitoS-tress Assay Protocol (Agilent). MitoStress test data were obtained using XF96 Seahorse Wave software, Agilent. Citrate Synthase Normalization Assay was per-formed on the cultured cell plate to normalize the measured Oxygen Consumption Rate (OCR) and ExtraCellular Acidification Rate (ECAR).

**Citrate Synthase Normalization Assay**. Citrate Synthase Normalization Assay was performed on Seahorse XF96 Cell Culture Microplate (Agilent) right after seahorse assay or on plates frozen at −80 °C with leftover medium aspirated. All leftover medium was removed and replaced with 113 µL of CS buffer (200 mM Tris buffer at pH8.0, 0.2% Triton X-100 (v/v), 100 µM DTNB (Sigma) in 100 mM Tris buffer at pH8.0, 1mM Acetyl-CoA (Sigma)) per well. In all, 5 µL of 10mM Oxa-loacetate (Sigma) in water was added to each well as reaction substrate. Absorbance at 412 nM at 37 °C was recorded with Tecan Microplate Reader M200 (Tecan) at minimal time interval for 8 min Citrate Synthase activity was then calculated using the formula below

$$\text{CS activity} = \sum_{i=1}^{n} \left[ \left( \frac{A_i - A_0}{t_i - t_0} \right) \right] / n \cdot \left( n \in N^* \right)$$

$n$ is the total number of absorbance records; $A$ is the absorbance recorded; $t$ is the time.

**Respiratory chain enzymatic assays**. Respiratory chain enzymatic assays were performed with zebrafish skeletal muscle homogenates according to the protocol as described by Spinazzi et al.[35].

**Statistics and reproducibility**. Unless otherwise stated, all data are presented as Mean ± standard error of mean (SEM.). Statistical approaches to test differences of means were performed using Prism 5.0. The statistical test used in each panel is indicated in each accompanying figure legend. Where *p*-values are not explicitly indicated, significance levels follow the convention of *$p < 0.05$, **$p < 0.01$, ***$p < 0.001$.

The reproducibility of representative data is as follows: Experiments of Figs. 1e, 2e, f, i–k, 3b, 4a, 5f, and 6f, Supplementary Figs. 1e, 2d, e, g–k, 4b, d, and 6f, h–k

were repeated independently for two times with similar data obtained in all replicates. Experiments of Figs. 2c, d, g, h, l, m, 3c, 4b, and 5e, Supplementary Figs. 2f, 3e, and 6a, e were repeated independently for three times with similar data obtained in all replicates. Experiments of Fig. 4e, Supplementary Figs. 4a and 6b were repeated independently for more than three times with similar data obtained in all replicates.

**Reporting summary**. Further information on research design is available in the Nature Research Reporting Summary linked to this article.

## Data availability
Proteomics data are available from Proteome Xchange. Accession: PXD016718, Source data for all western blots shown are available as a manuscript supplement,10.6084/m9. figshare.11406999, [https://figshare.com/s/76fbb5bf11af6d5d01af]. All reagents cited, including animals, will be available upon request through MTA.

Weblinks of publicly available datasets that used in the study: [http://www.sorfs.org/] Skeletal muscle dataset for GSEA: [https://www.ncbi.nlm.nih.gov/geo/query/acc.cgi? acc=GSE120862] Skin data for GSEA:
[https://www.ncbi.nlm.nih.gov/geo/query/acc.cgi?acc=GSE85861] Liver dataset for GSEA: [https://www.ncbi.nlm.nih.gov/geo/query/acc.cgi?acc=GSE94660] Heart dataset for GSEA: [https://www.ebi.ac.uk/ega/studies/EGAS00001002454]

## Code availability
All R code for WCGNA/GSEA analyses are available from [https://github.com/ LenaHoLab/Zhang-et-al-manuscript-R-code].

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

## Acknowledgements

We thank the IMCB Aquatics facility, the IMU Imaging facility and the AMPL histo-pathology unit at A*STAR for providing technical support; JP Kovalik and Ching Jianhong from the Duke-NUS metabolomics facility for metabolomic analysis including assay development; Hu Zhen, Kor Chia Yee, and Jes Kwek Hui Min for their technical support; Aida Moreno Moral and Cheryl Lee for their inputs on bioinformatics analyses; Brijesh Kumar Singh for his assistance with the Agilent Seahorse platform; Sudipto Roy from IMCB for help with zebrafish muscle analyses. We thank Pui Mun Wong and Serene Chng for their assistance in generating zebrafish crispr knockout, and Wei Leong Chew from Genome Institute of Singapore for assistance with AAV generation. We thank Radiance Lim for her assistance in AAV injection. We thank the Bio21 Melbourne Mass Spectrometry and Proteomics Facility (**MMSPF**) for the provision of instru-mentation, training, and technical support. D.A.S. is supported by Australian National Health & Medical Research Council (NHMRC) grants GNT1140851 and GNT1140906. L.L. and C.M. are supported by Fondation Ernst & Lucie Schmid-heiny. This work is funded by fellowships NRF-NRFF2017-05 (National Research Foundation of Singapore) and HHMI-IRSP55008732 (Howard Hughes Medical Institute International Research Scholar Program) awarded to L.H.

## Author contributions

L.H. and S.Z. conceptualized the study, designed, performed, and interpreted all experi-ments. S.Z. and G.F. carried out the SEP screen. C.L., J.H.P., C.T., R.K.C., S.J., J.C.F., and C.L.W. performed and interpreted specific experiments. B.K., V.O., G.M., and E.P. designed and performed computational analyses on sORF selection and mitochondrial functional prediction. S.N., P.S., B.Reversade, and L.S. provided reagents. R.M.S. and L.C. W. carried out proteomic analyses. C.M. carried out experiments and P.D.R. performed conservation analyses under the supervision of L.L.; N.S.J., and L.T.K. implemented MitoCore model. B.Reljić and D.A.S. performed proteomics and biochemical analyses and edited the manuscript. L.H. and S.Z. wrote the manuscript.

## Competing interests
The authors declare no competing interests.
