## [Peer Review File · Nature Communications]

Reviewers' comments:

Reviewer #1 (Remarks to the Author):

In this work Zhang et al. have performed an extensive and sophisticated computational analysis to predict the mitochondrial location and function of uncharacterized small open reading frame (sORF)-encoded peptides (SEPs). The mitochondrial location of several SEPs was validated by immunofluorescence, and the authors focused for further functional characterization on the protein encoded by C2orf73, which they renamed BRAWNIN (BR). The authors clearly demonstrate that BR is located in the mitochondrial inner membrane, and its loss affects respiratory chain function and bioenergetics. According to the authors, this effect mainly occurs at the level of respiratory chain complex III, and they propose BR functions as a novel CIII assembly factor. This work is very interesting and technically-sound. However, I have some concerns related to the interpretation and presentation of the functional data, and some technical aspects of this work must be improved in order to reach the right conclusions.

-From the data presented, I agree that the loss of BR influences mitochondrial bioenergetics probably at the respiratory chain level. What is less clear to me is the specific functional effect of BR on complex III (CIII) assembly. Although the most severe functional defects mainly affect CIII related activities and assembly in the KO zebrafish model, there is also a clear decrease in CIV subunits (Figs. 4k, 4l) and CIV activity measurements should be repeated to improve the error bar (Fig. S4j). The effect of BR deficiency on decreased CIV should be better explained (the fact that CIV is not integrated in supercomplexes has never been proven to be a reason for CIV destabilization, as most CIV actually accumulates as a free unit as a result of SCs destabilization). In addition, the decreased levels observed in CIII subunits, CIII and CIII-related activities are very mild in shRNA-cells, do not correlate with the decrease in BR levels and look similar to the decreased levels of other respiratory chain complexes, contrary to the authors' statements (Figs. 3h-j).

-In particular, complex IV should be included in the doxycycline experiments shown in Figure 3k, which should include statistics. This is relevant for the conclusions of this work.

-Complex II must also be included in Fig. 3k, since it is the only respiratory chain complex not affected by doxycycline treatment, and in Fig. 4k, as it is easier to validate as a mitochondrial loading control than an overall Coomassie staining.

-Error bars should be calculated for the controls (Src) presented in Fig. 3i.

-In page 3, ..."Moreover, BR protein abundance in mouse tissues correlates with mitochondrial content, being high in brown adipose, cardiac and skeletal muscle but virtually undetectable in white adipose tissue (Fig.S2e)". This experiment has not been done, so any direct correlation is based on theoretical issues. A clear measurement of the mitochondrial content in these specific samples should be provided.

-In page 5, ..."the interaction between BR and UQCRC1 detected in human cells was validated by co-IP/western blot in digitonin solubilized mouse heart mitochondria expressing a Br-FLAG construct (Fig. S3f). These data suggest that BR, an IMM peptide, directly interacts with the respiratory chain at CIII." To demonstrate this statement the co-IP with anti-BR must be extended to other CIII subunits. Reverse co-IP with anti-UQCRC1 should also be shown in fig S3f.

-The conclusion that some BR is secreted (Fig.S2f) is not clearly supported by the experimental data, as the authors show that BFA treatment did not work well on this protein. Probably the amount of "secreted" BR comes from cell lysis during experimental manipulation. How can the authors differentiate between these two possibilities?

-Indicate digitonin concentrations used in Fig. 2m.

-If the BR protein is finally proven to have a specific role on CIII assembly, it would be desirable to follow the accepted nomenclatures for human CIII assembly factors (UQCC4 or similar) rather than coming up with a totally arbitrary name like BRAWNIN.

-This work presents some experimental data performed on human skeletal muscle and heart. I am particularly concerned about the lack of an ethical statement. A written informed consent from the individuals whose samples were used for this study should be added.

Reviewer #2 (Remarks to the Author):

The manuscript "BRAWNIN: A sORF-encoded Peptide Essential for Vertebrate Mitochondrial Complex III Assembly" by S. Zhang reports the identification of the mitochondrial peptide Brawnin as a new important factor for respiratory chain activity. The authors identified Brawnin as a peptide that resides in the inner membrane of mitochondria. In line with its mitochondrial localization, depletion of Brawnin either via si- or shRNAs in mammalian cell lines or via a stable KO in zebrafish results in a decrease in the rate of oxygen consumption. Co-immunoprecipitation experiments revealed that Brawnin interacts with the Complex III subunit Uqcrc1. Moreover, lack of Brawnin leads to impaired assembly of Complex III and decreases Uqcrc1 levels in mitochondria. Overall, the paper is well written, and the figures are generally well presented (exceptions are commented on below). While the data is overall of high quality, some results seem contradictory, which weakens the overall very interesting paper about a timely topic of high relevance also to a non-specialized audience.

Main concerns:

1. There appears to be a much weaker phenotype in tissue culture knockdown experiments than in zebrafish KOs, and a discrepancy between the almost complete loss of Complex III activity in zebrafish KOs despite. This could be due to the different KD approaches (KD versus KO), yet it is unclear why the authors did not generate a KO in tissue culture lines to have a clean phenotype for direct molecular characterization. The different levels of severity observed upon siRNA and shRNA knockdown in cell culture (the assembled Complex III is decreased by only 30 %) versus knockout in zebrafish (94% reduction of assembled Complex III) questions what the primary phenotype of lack of Brawnin is, and how much of the severe phenotype in zebrafish (almost complete loss of Complex III) might be a more long-term effect from sub-optimal assembly of Complex III. To reconcile these differences in severities and knockdown approaches, it would be important to assess whether KO of Brawnin in cell culture will recapitulate the complete loss of Complex III observed in zebrafish KOs. Furthermore, it is unclear how fish with 94% reduction of Complex III can survive until 40 days post fertilization, while Uqcrc1 knockout in mice has been reported to cause early embryonic death (W. Shan et al. Critical role of Uqcrc1 in embryo survival, brain ischemic tolerance and normal cognition in mice, 2019).
2. Brawnin's absence causes a drastic decrease in assembled Complex III in knockout zebrafish. However, at the same time enzymatic activity of Complex III is not significantly affected in knockout fish. These results contradict each other. I would suggest measuring enzymatic activity of isolated Complex III (if it is not possible in zebrafish, then in knockout human cell culture) and electron transport activity of individual Complex III.
3. Co-immunoprecipitation experiments revealed that Brawnin interacts with the Complex III subunit Uqcrc1. However, Uqcrc1 band is also visible in the control (yet in a reduced amount). To confirm this interaction, I would suggest performing reciprocal co-immunoprecipitation.
4. The authors have not addressed whether Brawnin is important for Uqcrc1 stabilization or proper assembly in Complex III. Does Brawnin's depletion specifically affect Uqcrc1 (e.g. stabilizing specifically Uqcrc1), or more generally Complex III assembly? Are other subunits of the Complex III present in the Brawnin KO, or equally destabilized as Uqcrc1? A MS experiment would

provide a direct answer, and would also allow to assess the protein levels for Complex III subunits (particularly Uqcrc1) in the mitochondrial fraction versus whole cell lysate.

5. And relatedly To which extent is the Brawnin KO pt due to destabilization of Uqcrc1? E.g. can the KO phenotype in zebrafish be rescued by reintroduction of 1) Brawnin (e.g. mRNA injection in MZs or transgenic rescue should result in a reduced severity or rescue of the decreased assembly of Complex III); and 2) Uqcrc1? Since Uqcrc1 levels are reduced upon Brawnin depletion, Brawnin's main function could be to stabilize Uqcrc1 (or alternatively, it might play a more general role in the assembly of the Complex III).

Minor points that would further improve the manuscript:

1. The authors propose that Brawnin is under AMPK-PGC1 α control. While this is shown rather convincingly for the AMPK pathway (Fig. 3b) since 24 hours after AICAR treatment the expression of Brawnin and other OXPHOS subunits increases, Brawnin mRNA levels only slightly increase one day after overexpression of PGC1 α (Fig. S3b). To make the claim that Brawnin is regulated by PGC1 α more convincing data and positive controls for PGC1 α regulated proteins would need to be shown.

2. Sentence in the Abstract "Using a mitochondrial prediction and validation pipeline for small open-reading-frame (sORF)-encoded peptides (SEPs), we report the discovery of 16 endogenous mitochondrial SEPs (mito-SEPs) associated with oxidative phosphorylation (OXPHOS)". Elaborate a bit more, what do you mean by "associated".

3. Main text page 3 "Of the 16 new mito-SEPs, the expression of 8 correlated with respiratory chain and electron transport function above the 90th percentile (Fig. 1e), suggesting that half of them participate directly in oxidative phosphorylation". Good correlation in expression level does not suggest that these peptides participate directly in oxidative phosphorylation. Mitoregulin (MtlN) is a vivid example of this.

4. Main text page 4 "Indeed, BR depletion significantly decreased basal and maximal respiration, spare capacity and ATP production in both siBR and shBR U87MG (Fig. 3d-f, S3c)". Brawnin depletion does not lead to significant decrease in ATP production according to your results. Please, correct this sentence.

5. Main text page 7 "Furthermore, spectrophotometric measurement of RC enzymatic activities revealed that all CIII-dependent activities were indeed impaired in zKO muscle homogenates (Fig. S4j)". Not all CIII-dependent activities are impaired.

6. Figure 1a,b. Please plot densities in a semi-transparent mode that overlaying densities will all be visible. What would help to bring the point better across is to give the percentages of genes (total versus mitochondria) that are <100 aa (e.g. amongst all proteins there might be only <1 % below 100aa, but within mitochondrial proteins it might be >5%).

7. Figure 1e and several other Figure legends. Please, explain the data shown in the picture in more detail in figure legends and include here a peptide that is expressed, but not localized in mitochondria.

8. Figure 2a. Transfer the upper part to supplementary and indicate what lines with different color mean.

9. Figure 2b. Please indicate peptide epitope detected by antibodies on amino acid sequence scheme.

10. Figure 2e. Please indicate whether you used synthetic peptide or Brawnin without a transmembrane domain (the label does not fit the figure legend).

11. Figure 2h-k. Control is required. Please show overlap with characteristic mitochondrial structure using Tom20 antibodies.

12. Figure 2m,n. Please indicate in the Figure directly which proteins are outer versus inner mitochondrial proteins.

13. Figure 3a. Please introduce a positive control.

14. Figure 3d,e. Please indicate what the different lines and colors correspond to.

15. Figure 3j. Please calculate the relative abundance of these proteins.

16. Figures 4c-o. Please, indicate everywhere whether you used homozygous knockout or maternal zygotic knockout.

17. Figure S2d. Please, transfer it to the main pictures, as a control to Figure 2f,g.
18. Figure S2e. Control for mitochondrial content is required.

Reviewer #3 (Remarks to the Author):

These are excellent, highly informative studies reporting sORF-encoded peptides, in particular, C12orf73 is crucial for Complex III assembly, possibly constitutes its activity. It is regulated by AMPK-dependent mechanisms. They went on tested whether ablation of BR impacts zebrafish development and growth. They have observed a remarkable phenotype lactic acidosis and early on-set of death. Although the manuscript is neatly crafted, mitochondrial complex activities and AMPK status need to be examined.

As we all know, in recent years most studies show OCR measurement and extrapolate the complex activities.

Since the KO animals show higher levels of complex II substrate (succinate), authors must measure individual complex activities especially CI, CII and CIII (Hawkins B.J. et al 285(34):26494-505. Journal of Biological chemistry 2010).

Having observed elevation of lactate and possibly higher AMP levels in KO fish, it is important to measure phosphorylated state of AMPK levels. A recent study from Tomar et al revealed that higher levels of AMP failed to phosphorylate AMPK due to increased activation of PP4 activity through cytosolic Ca²⁺ rise (Tomar D. et al., 2019 Cell Reports 26(13):3709-3725.e7). This work needs to be discussed and interpret the authors' findings.

Point-by-point response

Reviewer #1 (Remarks to the Author):

In this work Zhang et al. have performed an extensive and sophisticated computational analysis to predict the mitochondrial location and function of uncharacterized small open reading frame (sORF)-encoded peptides (SEPs). The mitochondrial location of several SEPs was validated by immunofluorescence, and the authors focused for further functional characterization on the protein encoded by *C2orf73*, which they renamed BRAWNIN (BR). The authors clearly demonstrate that BR is located in the mitochondrial inner membrane, and its loss affects respiratory chain function and bioenergetics. According to the authors, this effect mainly occurs at the level of respiratory chain complex III, and they propose BR functions as a novel CIII assembly factor. This work is very interesting and technically-sound. However, I have some concerns related to the interpretation and presentation of the functional data, and some technical aspects of this work must be improved in order to reach the right conclusions.

-From the data presented, I agree that the loss of BR influences mitochondrial bioenergetics probably at the respiratory chain level. What is less clear to me is the specific functional effect of BR on complex III (CIII) assembly. Although the most severe functional defects mainly affect CIII related activities and assembly in the KO zebrafish model, there is also a clear decrease in CIV subunits (Figs. 4k, 4l) and CIV activity measurements should be repeated to improve the error bar (Fig. S4j). The effect of BR deficiency on decreased CIV should be better explained (the fact that CIV is not integrated in supercomplexes has never been proven to be a reason for CIV destabilization, as most CIV actually accumulates as a free unit as a result of SCs destabilization).

Thank you for pointing out the apparent CIV defect. We have carefully expanded our examination of potential CIII and CIV defects in BR knockout zebrafish. We conclude that Br deletion results specifically in CIII but not CIV loss. These conclusions are supported by the following data:

- 1) CIV subunits are not downregulated, as measured by quantitative proteomics of purified mitochondria. In contrast, almost all detected CIII components are downregulated with 5 passing significance threshold (Fig. 6a-c).
- 2) Confirming our proteomics data, SDS-PAGE analysis of an expanded set of animals and ETC proteins indicate that CIV subunits MTCOI and COX4 are not downregulated (Figures 6d-e). The original data showing CIV downregulation in one animal (original Fig. 4l) was due to variation across animals.
- 3) Monomeric CIV complex levels, as measured by BN-PAGE of zebrafish KO (Fig. 6f), human KD (Fig. S6b-d) and mouse KO (Fig. S6h) mitochondria indicate that that CIV complex as a whole is not destabilized.
- 4) Improved RCA measurements (Fig. 6g) demonstrate clear and significant impairment in all CIII containing activity (CIII, CII+III, CI+III). CIV enzymatic activity, as measured using reduced cytochrome c as an electron donor, is not significantly affected.
- 5) However, electron flow through CIV to oxygen measured using TMPD as an electron donor is severely attenuated (original Fig. 4m). This functional defect could potentially be explained by deficits in cytochrome c as an electron carrier secondary to CIII loss. Because this is still under

investigation, we have decided to remove Fig. 4m in the revised submission, until we can clarify the underlying cause of the observed functional defect in CIV.

In addition, the decreased levels observed in CIII subunits, CIII and CIII-related activities are very mild in shRNA-cells, do not correlate with the decrease in BR levels and look similar to the decreased levels of other respiratory chain complexes, contrary to the authors' statements (Figs. 3h-j).

The mild phenotype observed in U87MG shBR cells compared to that in zebrafish muscle can potentially be attributed to differences in BR requirement in different tissues (skeletal muscle vs epithelial cells), acute versus chronic depletion (cell culture vs animal KO), knockdown versus knockout, or evolution of redundant mechanisms safeguarding CIII stability/assembly in higher vertebrates.

To verify the requirement for BR in CIII assembly in another mammalian model, we have now generated a mouse embryonic fibroblast *Br* knockout cell line that shows a clear reduction of CIII dimer, with no other discernible defects in ETC complexes (Fig. S6e-i). The primary defect of BR depletion is therefore a partial loss of CIII that is likely to be progressively amplified, resulting in near complete loss of CIII in the adult zebrafish knockout. This is reminiscent of murine mutants of CIII assembly factor BCS1L, which develop progressive loss of mature CIII owing to accumulated defects in RISP incorporation during the post-natal period (Levéen et al. 2011).

-In particular, complex IV should be included in the doxycycline experiments shown in Figure 3k, which should include statistics. This is relevant for the conclusions of this work.

-Complex II must also be included in Fig. 3k, since it is the only respiratory chain complex not affected by doxycycline treatment, and in Fig. 4k, as it is easier to validate as a mitochondrial loading control than an overall Coomassie staining.

Thank you for the suggestion to do so. We have updated this figure to include SDHA (CII) and COX4 (CIV) from the same blot to show equal loading and normal free CIV levels. We have also performed quantitation of triplicate runs to provide statistics. To accommodate our expanded proteomic and biochemical analyses of zebrafish KOs, Fig. 3K has been moved to Fig. S6d.

Fig. 4k (now Fig. 6f) has now been expanded to include markers for Complex I - IV for which zebrafish cross-reactive antibodies were available.

-Error bars should be calculated for the controls (Scr) presented in Fig. 3i.

Fig. 3i is now Fig. S6c. Error bars are from 5 separate BN-PAGE experiments compared to 3 in the first submission. In each experiment, the relative intensity of the band-in-question in shBR is calculated as a percentage of the corresponding band in the Scr control, which is set at 100%. This explains why there are no error bars for the Scr.

-In page 3, ...”Moreover, BR protein abundance in mouse tissues correlates with mitochondrial content, being high in brown adipose, cardiac and skeletal muscle but virtually undetectable in white adipose tissue (Fig.S2e)”. This experiment has not been done, so any direct correlation is based on theoretical issues. A clear measurement of the mitochondrial content in these specific samples should be provided.

Fig. S2e (now Fig. S2f) has been re-probed with ETC complexes. Indeed, BR levels co-relate with that of ETC complexes and by inference, mitochondrial abundance.

-In page 5, “the interaction between BR and UQCRC1 detected in human cells was validated by co-IP/western blot in digitonin solubilized mouse heart mitochondria expressing a Br-FLAG construct (Fig. S3f). These data suggest that BR, an IMM peptide, directly interacts with the respiratory chain at CIII.” To demonstrate this statement the co-IP with anti-BR must be extended to other CIII subunits. Reverse co-IP with anti-UQCRC1 should also be shown in fig S3f.

We have extended the co-IP with BR to other CIII subunits, namely MT-CYB, UQCRFS1 and UQCRH (Fig. 5e), demonstrating that the interaction of BR with CIII is not limited to UQCRC1. Reverse co-IP was difficult because available CIII subunit antibodies did not work for CIII IP in digitonin-solubilized mitochondria (i.e. anti-MT-CYB, UQCRC1, UQCRC2, UQCRH, UQCRFS1). We therefore performed the reverse co-IP experiments using UQCRQ-FLAG and BR-HA constructs in HEK293T mitochondrial lysates and detected interaction between these two proteins (Fig. 5f). UQCRQ was chosen as the C-terminus of this subunit is surface-exposed (i.e. not within a hydrophobic membrane-buried domain) and hence more amenable to immunoprecipitation.

-The conclusion that some BR is secreted (Fig.S2f) is not clearly supported by the experimental data, as the authors show that BFA treatment did not work well on this protein. Probably the amount of “secreted” BR comes from cell lysis during experimental manipulation. How can the authors differentiate between these two possibilities?

Under the conditions used for Fig. S2f (now Fig. S2g), there is no non-specific cell lysis, as evidenced by the ability of BFA to completely block the secretion of C4ORF48. We conclude that under artificial conditions of overexpression, some BR can be “secreted”, but we do not think this is bona fide secretion due to the inability to be blocked by BFA (Fig. S2g) and the lack of signal peptide cleavage (Fig. S2g,h). This is to address the point that the N-terminal sequence is a mitochondrial targeting sequence, rather than a signal peptide. We have rephrased the text to clarify this point, that we in fact agree with you that this is not real secretion.

-Indicate digitonin concentrations used in Fig. 2m.

We have revised Fig. 2m (now Fig. 2l) to indicate digitonin concentrations.

-If the BR protein is finally proven to have a specific role on CIII assembly, it would be desirable to follow the accepted nomenclatures for human CIII assembly factors (UQCC4 or similar) rather than coming up with a totally arbitrary name like BRAWNIN.

BRAWNIN is an abbreviation for *Cytochrome BC₁ - Respirasome Assembly Affected When INsufficient*. We will consider proposing UQCC4 as the official name once we have definitive proof that BR functions directly as an accessorial factor for CIII assembly and we thank you for this suggestion.

-This work presents some experimental data performed on human skeletal muscle and heart. I am particularly concerned about the lack of an ethical statement. A written informed consent from the individuals whose samples were used for this study should be added.

We appreciate your caution. We purchased tissue slides from US Biomax, which states on their website that “All human tissues are collected under HIPPA approved protocols.” (<https://www.biomax.us/FAQs>). We have indicated this in the methods section. Under HBRA regulations, de-identified FFPE samples are not considered human tissue and does not require that the researcher (i.e. us) maintain copies of the written consent from the donors since we are not the tissue bank. Of note, Fig. 2h-i from the original submission has been moved to Fig. S2e in the revised submission due to space constraints.

Reviewer #2 (Remarks to the Author):

The manuscript “BRAWNIN: A sORF-encoded Peptide Essential for Vertebrate Mitochondrial Complex III Assembly” by S. Zhang reports the identification of the mitochondrial peptide Brawnin as a new important factor for respiratory chain activity. The authors identified Brawnin as a peptide that resides in the inner membrane of mitochondria. In line with its mitochondrial localization, depletion of Brawnin either via si- or shRNAs in mammalian cell lines or via a stable KO in zebrafish results in a decrease in the rate of oxygen consumption. Co-immunoprecipitation experiments revealed that Brawnin interacts with the Complex III subunit Uqcrc1. Moreover, lack of Brawnin leads to impaired assembly of Complex III and decreases Uqcrc1 levels in mitochondria. Overall, the paper is well written, and the figures are generally well presented (exceptions are commented on below). While the data is overall of high quality, some results seem contradictory, which weakens the overall very interesting paper about a timely topic of high relevance also to a non-specialized audience.

Main concerns:

1. There appears to be a much weaker phenotype in tissue culture knockdown experiments than in zebrafish KOs, and a discrepancy between the almost complete loss of Complex III activity in zebrafish despite. This could be due to the different KD approaches (KD versus KO), yet it is unclear why the authors did not generate a KO in tissue culture lines to have a clean phenotype for direct molecular characterization. The different levels of severity observed upon siRNA and shRNA knockdown in cell culture (the assembled Complex III is decreased by only 30 %) versus knockout in zebrafish (94% reduction of assembled Complex III) questions what the primary phenotype of lack of Brawnin is, and how much of the severe phenotype in zebrafish (almost complete loss of Complex III) might be a more long-term effect from sub-optimal assembly of Complex III. To reconcile these differences in severities and knockdown approaches, it would be important to assess whether KO of Brawnin in cell culture will recapitulate the complete loss of Complex III observed in zebrafish KOs. Furthermore, it is unclear how fish with 94% reduction of Complex III can survive until 40 days post fertilization, while Uqcrc1 knockout in mice has been reported to cause early embryonic death (W. Shan et al. Critical role of Uqcrc1 in embryo survival, brain ischemic tolerance and normal cognition in mice, 2019).

We thank you for your critical and careful review. To verify the requirement for BR in CIII assembly in a cell line, we generated a mouse embryonic fibroblast (MEF) KO cell line, which shows a clear reduction of CIII dimers with no discernible defect in other ETC complexes (Fig. S6e-h). Hence, acute depletion of BR in human cell lines by both shRNA and Crispr-mediated KO in a mouse cell line both indicate that the proximal effect of BR loss is the reduction in mature assembled CIII (Fig. S6b-d, e-h), arguing that this is the primary defect.

However, in contrast to the near complete reduction in zebrafish, CIII reductions observed in both U87MG and MEF are indeed only partial. This suggests that the discrepancy is not simply due to KO vs KD. Rather, we favor the explanation that sub-optimal CIII assembly in BR mutants progressively amplifies due to long-term BR loss. This is reminiscent of the gradual loss of mature CIII in *Bcs1L* mutants owing to initially mild defects in RISP incorporation that gradually worsen during the post-natal period (Levéen et al. 2011).

Alternatively, BR might have functional paralogues in higher vertebrates not present in zebrafish that compensate for its loss in mouse and human cultured cells, and we are currently investigating this possibility.

While knockouts of core subunits of CIII are embryonic lethal in the mouse (e.g. *Uqcrc1*), knockouts of CIII accessory factors (e.g. *TTC19*) result in milder, post-natal onset progressive phenotypes (Bottani et al. 2017). Likewise, while mutations in structural CIII subunits are extremely rare (Gaignard et al. 2017) mutations in accessory subunits like *TTC19*, *LYRM7* and *BCS1L* are more frequent and present with phenotypes with variable onset and severity (Ghezzi and Zeviani 2012). Put together, these observations suggest that BR is an accessory factor and not a core subunit of CIII. In line with this, C12ORF73 (BR) has never been observed in solved structures or mass spectrometric analyses of Complex III (our unpublished observation).

2. Brawnin's absence causes a drastic decrease in assembled Complex III in knockout zebrafish. However, at the same time enzymatic activity of Complex III is not significantly affected in knockout fish. These results contradict each other. I would suggest measuring enzymatic activity of isolated Complex III (if it is not possible in zebrafish, then in knockout human cell culture) and electron transport activity of individual Complex III.

We have analyzed further 8 animals per genotype using respiratory chain assays (RCAs) to enable more robust statistical quantification. These new data are presented in Fig. 6g. Indeed, only CIII, CII+III and CI+III RCAs were significantly different between WT and KO. CI, CII and CIV activities were not significantly reduced.

Altogether, 12 WT and 11 KO animals were analyzed across 3 experiments. The average CS-normalized value for the WTs in each experiment was set at 100%, and the relative activity of each animal was calculated as a percentage of that reference value. These assays demonstrate clear and significant impairment in CIII enzymatic activity in isolated CIII (UbH₂ :: oxidised Cyt c) and electron flow from CI to CIII (NADH :: oxidised Cyt c) and from CII to CIII (succinate :: oxidised Cyt c). Since there are no significant defects in CI and CII enzymatic activity (Fig. 6g) or protein levels (Fig. 6a-c), the electron flow defects seen in CI+III and CII+III are due to CIII defects. Furthermore, exogenous ubiquinol (UbH₂) (used as electron donor in the CIII RCA) did not suppress CIII defects in the CIII RC assay (Fig. 6g), arguing against Coenzyme Q deficiency as a cause of CI+III and CII+III defects.

3. Co-immunoprecipitation experiments revealed that Brawnin interacts with the Complex III subunit *Uqcrl1*. However, *Uqcrl1* band is also visible in the control (yet in a reduced amount). To confirm this interaction, I would suggest performing reciprocal co-immunoprecipitation.

We have repeated and extended the co-IP of BR with other CIII subunits, namely MT-CYB, UQCRFS1 and UQCRH (Fig. 5e), demonstrating that the interaction of BR with CIII is not limited to UQCRC1. Reverse co-IP was difficult because available CIII subunit antibodies did not work for CIII IP in digitonin-solubilized mitochondria (i.e. anti-MT-CYB, UQCRC1, UQCRC2, UQCRH, UQCRFS1, UQCRB,

UQCR10), making co-IP impossible. We therefore performed the reverse co-IP experiments using UQCRQ-FLAG and BR-HA constructs in HEK293T mitochondrial lysates and detected interaction between these two proteins (Fig. 5f). UQCRQ was chosen as the C-terminus of this subunit is surface-exposed (i.e. not within a hydrophobic membrane-buried domain) and hence more amenable to immunoprecipitation.

4. The authors have not addressed whether Brawnin is important for Uqcr1 stabilization or proper assembly in Complex III. Does Brawnin's depletion specifically affect Uqcr1 (e.g. stabilizing specifically Uqcr1), or more generally Complex III assembly? Are other subunits of the Complex III present in the Brawnin KO, or equally destabilized as Uqcr1? A MS experiment would provide a direct answer, and would also allow to assess the protein levels for Complex III subunits (particularly Uqcr1) in the mitochondrial fraction versus whole cell lysate.

MS analyses of *zf br* KO mitochondria (Fig. 6a) and MEF KO mitochondria (Fig. S6g) point to a global and specific destabilization of CIII complex that is not restricted to Uqcr1. Whether this is due to reduced production or stability of CIII subunits, or reduced CIII assembly leading to degradation of unincorporated subunits is the subject of ongoing investigation. We are working on a comprehensive study of how BR mechanistically enables CIII assembly and hope to present it in a future manuscript.

5. And relatedly to which extent is the Brawnin KO pt due to destabilization of Uqcr1? E.g. can the KO phenotype in zebrafish be rescued by reintroduction of 1) Brawnin (e.g. mRNA injection in MZs or transgenic rescue should result in a reduced severity or rescue of the decreased assembly of Complex III); and 2) Uqcr1? Since Uqcr1 levels are reduced upon Brawnin depletion, Brawnin's main function could be to stabilize Uqcr1 (or alternatively, it might play a more general role in the assembly of the Complex III).

Our data suggest that the *br* KO phenotype is due to mitochondrial deficiency secondary to near complete loss of CIII. Our working hypothesis is that Br works early in the assembly of CIII. Since Uqcr1 is not the only CIII component lost in *br* KOs, we do not predict that *uqcr1* reintroduction alone will be sufficient to rescue the whole organism phenotype of *br* KOs.

Minor points that would further improve the manuscript:

1. The authors propose that Brawnin is under AMPK-PGC1a control. While this is shown rather convincingly for the AMPK pathway (Fig. 3b) since 24 hours after AICAR treatment the expression of Brawnin and other OXPHOS subunits increases, Brawnin mRNA levels only slightly increase one day after overexpression of PGC1a (Fig. S3b). To make the claim that Brawnin is regulated by PGC1a more convincing data and positive controls for PGC1a regulated proteins would need to be shown.

We have included positive controls in Fig. S3b indicating that PGC1a increases *Br* transcript levels along with that of other ETC components. By contrast, a structural component of the mitochondria, *Tomm20* is not changed.

2. Sentence in the Abstract “Using a mitochondrial prediction and validation pipeline for small open-reading-frame (sORF)-encoded peptides (SEPs), we report the discovery of 16 endogenous mitochondrial SEPs (mito-SEPs) associated with oxidative phosphorylation (OXPHOS)”. Elaborate a bit more, what do you mean by “associated”.

The association we report here is based on co-expression at the transcript level. Cell types that express higher levels of genes encoding electron transport components and genes involved in oxidative respiration also express higher levels of BR and the mito-SEPs highlighted in red in Fig. 1e. To avoid misleading the reader, we have changed the sentence in the abstract to:

“Using a mitochondrial prediction and validation pipeline for small open-reading-frame (sORF)-encoded peptides (SEPs), we report the discovery of 16 endogenous nuclear encoded, mitochondrial-localized SEPs (mito-SEPs)”.

3. Main text page 3 “Of the 16 new mito-SEPs, the expression of 8 correlated with respiratory chain and electron transport function above the 90th percentile (Fig. 1e), suggesting that half of them participate directly in oxidative phosphorylation”. Good correlation in expression level does not suggest that these peptides participate directly in oxidative phosphorylation. Mitoregulin (MtlN) is a vivid example of this.

We agree with you and have changed the sentence to *“Of the 16 new mito-SEPs, the expression of 8 correlated with respiratory chain and electron transport function above the 90th percentile (Fig. 1e), suggesting that half of them participate directly **or indirectly** in oxidative phosphorylation”.* This is in agreement with MtlN, which regulates oxidative phosphorylation indirectly by controlling the upstream process of beta-oxidation.

4. Main text page 4 “Indeed, BR depletion significantly decreased basal and maximal respiration, spare capacity and ATP production in both siBR and shBR U87MG (Fig. 3d-f, S3c)”. Brawnin depletion does not lead to significant decrease in ATP production according to your results. Please, correct this sentence.

We have changed the sentence to “Indeed, BR depletion significantly decreased basal, maximal respiration, spare capacity and reduced ATP production in both siBR and shBR U87MG (Fig. 3d-f, S3c)”

5. Main text page 7 “Furthermore, spectrophotometric measurement of RC enzymatic activities revealed that all CIII-dependent activities were indeed impaired in zKO muscle homogenates (Fig. S4j)”. Not all CIII-dependent activities are impaired.

Please see response to Main Concern Point #2. All CIII-dependent activities are significantly impaired (Fig. 6g).

6. Figure 1a,b. Please plot densities in a semi-transparent mode that overlaying densities will all be visible. What would help to bring the point better across is to give the percentages of genes (total versus

mitochondria) that are <100 aa (e.g. amongst all proteins there might be only <1 % below 100aa, but within mitochondrial proteins it might be >5%).

We have incorporated your helpful suggestions in the new Fig. 1a-c.

7. Figure 1e and several other Figure legends. Please, explain the data shown in the picture in more detail in figure legends and include here a peptide that is expressed, but not localized in mitochondria.

We have expanded the Figure 1e's legend (now Fig. 1f) and inserted the position of a non-mito-SEP i.e. expressed but not localized to the mitochondria.

8. Figure 2a. Transfer the upper part to supplementary and indicate what lines with different color mean.

We have moved the upper part of the figure to Fig S2a as it does not impact the interpretation of the data. This was to make room for point #17. The lines with different color indicate the RPF (riboseq count) of different human cell lines in sORFs.org. We have updated the figure legend to indicate this.

9. Figure 2b. Please indicate peptide epitope detected by antibodies on amino acid sequence scheme.

We have indicated the position of the antigens used to raise the BR antibodies used in this study.

10. Figure 2e. Please indicate whether you used synthetic peptide or Brawnin without a transmembrane domain (the label does not fit the figure legend).

We used a synthetic peptide lacking the transmembrane domain. We have rectified the erroneous figure legend. Thank you for spotting this.

11. Figure 2h-k. Control is required. Please show overlap with characteristic mitochondrial structure using Tom20 antibodies.

We now provide immunofluorescence images of WT mouse skeletal muscle co-stained with α -BR and α -Citrate Synthase (CS) (Fig. 2i) and Mito-Dendra2 transgenic mouse skeletal muscle (where mitochondria are epifluorescently labeled by GFP) stained with α -BR (Fig. 2j). Both demonstrate clear colocalization of BR with mitochondria.

12. Figure 2m,n. Please indicate in the Figure directly which proteins are outer versus inner mitochondrial proteins.

Please find the updated figures in panels 3l and 3m.

13. Figure 3a. Please introduce a positive control.

We have included the profile of *ATP5H*, a component of ATP Synthase as a positive control in Fig. 3a.

14. Figure 3d,e. Please indicate what the different lines and colors correspond to.

We apologize for the omission and have appended the legends to Fig. 3d and 3e.

15. Figure 3j. Please calculate the relative abundance of these proteins.

We have now provided quantitation for Fig. 3j, and have moved it to Fig. S6a.

16. Figures 4c-o. Please, indicate everywhere whether you used homozygous knockout or maternal zygotic knockout.

In general, adult animals used were zygotic KOs and experiments involving larvae employed maternal zygotic KOs.

Zygotic KOs are referred to as “zKO” and were used in Fig. 4 b,c,d,e,g, Fig. S4d, Fig. 6a-g, and Fig. S6j,k.

Maternal-zygotic KOs are referred to as “mzKO” and were used in Fig. 4 d,f, h, Fig. 5a,b and Fig.

S4b,c,e,f,g.

17. Figure S2d. Please, transfer it to the main pictures, as a control to Figure 2f,g.

Fig. S2d is now Fig. 2e.

18. Figure S2e. Control for mitochondrial content is required.

Fig. S2e (now Fig. S2f) has been re-probed with ETC complexes as a measure of mitochondrial content.

Indeed, BR levels co-relate with that of known ETC complexes.

Reviewer #3 (Remarks to the Author):

These are excellent, highly informative studies reporting sORF-encoded peptides, in particular, C12orf73 is crucial for Complex III assembly, possibly constitutes its activity. It is regulated by AMPK-dependent mechanisms. They went on tested whether ablation of BR impacts zebrafish development and growth. They have observed a remarkable phenotype lactic acidosis and early on-set of death. Although the manuscript is neatly crafted, mitochondrial complex activities and AMPK status need to be examined.

As we all know, in recent years most studies show OCR measurement and extrapolate the complex activities.

Since the KO animals show higher levels of complex II substrate (succinate), authors must measure individual complex activities especially CI, CII and CIII (Hawkins B.J. et al 285(34):26494-505. Journal of Biological chemistry 2010).

Thank you for your suggestion. We have carefully performed RC assays on isolated skeletal muscle mitochondria from 12 WT and 11 KO animals. These data are presented in Fig. 6g. Indeed, only CIII, CII+III and CI+III RCAs were significantly different between WT and KO. CI, CII and CIV activities were not significantly reduced.

These assays demonstrate clear and significant impairment in CIII enzymatic activity in isolated CIII (UbH₂ :: oxidised Cyt c) and electron flow from CI to CIII (NADH :: oxidised Cyt c) and from CII to CIII (succinate :: oxidised Cyt c). Since there are no significant defects in CI and CII enzymatic activity (Fig. 6g) or protein levels (Fig. 6a-c), the electron flow defects seen in CI+III and CII+III are due to CIII defects. Furthermore, exogenous ubiquinol (UbH₂) did not suppress CIII defects in the CIII RC assay (Fig. 6g) , arguing against Coenzyme Q deficiency as a cause of CI+III and CII+III defects.

Having observed elevation of lactate and possibly higher AMP levels in KO fish, it is important to measure phosphorylated state of AMPK levels. A recent study from Tomar et al revealed that higher levels of AMP failed to phosphorylate AMPK due to increased activation of PP4 activity through cytosolic Ca²⁺ rise (Tomar D. et al., 2019 Cell Reports 26(13):3709-3725.e7). This work needs to be discussed and interpret the authors' findings.

We attempted to measure pAMPK and pAcetyl CoA Carboxylase (pACC) levels in WT and KO zebrafish and were unfortunately unable to obtain cross-reactive antibodies. We therefore measured pAMPK and pACC status in Br KO mouse embryonic fibroblasts (MEF) and found that BR depletion led to increased activation of the AMPK pathway (Fig. S6i). This is consistent with lactate elevation and potentially higher AMP levels when Br is deficient, as you pointed out.

[REDACTED]

References:

Bottani, Emanuela, et al. "TTC19 plays a husbandry role on UQCRC1 turnover in the biogenesis of mitochondrial respiratory complex III." *Molecular cell* 67.1 (2017): 96-105.

Gaignard, Pauline, et al. "UQCRC2 mutation in a patient with mitochondrial complex III deficiency causing recurrent liver failure, lactic acidosis and hypoglycemia." *Journal of human genetics* 62.7 (2017): 729.

Ghezzi, Daniele, and Massimo Zeviani. "Assembly factors of human mitochondrial respiratory chain complexes: physiology and pathophysiology." *Mitochondrial Oxidative Phosphorylation*. Springer, New York, NY, 2012. 65-106.

Levéen, Per, et al. "The GRACILE mutation introduced into Bes11 causes postnatal complex III deficiency: a viable mouse model for mitochondrial hepatopathy." *Hepatology* 53.2 (2011): 437-447.

REVIEWERS' COMMENTS:

Reviewer #1 (Remarks to the Author):

The authors have successfully addressed all my concerns in the revised manuscript and robust conclusions can be now drawn from the data. I consider this to be an excellent work of general interest for biochemical research groups involved in the mechanistic models of mitochondrial biogenesis and metabolism, and the new results add a relevant new value to previous research articles. Therefore, I recommend its acceptance for publication.

Reviewer #2 (Remarks to the Author):

The authors have significantly improved their manuscript in the revised form by providing additional experimental data (KO cell line, additional RCA measurements in zebrafish revealing a more general defect in CIII, extended co-IP experiments) and increasing the clarity of Figures and Figure legends. With these improvements, my main concerns have been addressed.

The only point that has not been addressed is the suggested rescue experiment with BR expression in zebrafish embryos, yet given the overall convincing and strong data that the defect is indeed due to loss of BR the manuscript is already convincing in its current form.

One minor suggestion to incorporate in the discussion or text: The authors currently favor the idea that loss of BR leads to a defect in CIII assembly – yet it could also be that import of CIII subunits is impaired, which could in turn lead to a defect in assembly. Since the current data does not allow to distinguish between these two possibilities (and experiments to distinguish between these possibilities will go beyond the scope of this manuscript), I would suggest mentioning this explicitly as a possibility for the primary function of BR.

Reviewer #3 (Remarks to the Author):

The authors have adequately addressed all comments raised by the reviewer and the revised manuscript is appropriate for publication.

Point by point response to referee

REVIEWERS' COMMENTS:

Reviewer #1 (Remarks to the Author):

The authors have successfully addressed all my concerns in the revised manuscript and robust conclusions can be now drawn from the data. I consider this to be an excellent work of general interest for biochemical research groups involved in the mechanistic models of mitochondrial biogenesis and metabolism, and the new results add a relevant new value to previous research articles. Therefore, I recommend its acceptance for publication.

Thank you for the recommendation.

Reviewer #2 (Remarks to the Author):

The authors have significantly improved their manuscript in the revised form by providing additional experimental data (KO cell line, additional RCA measurements in zebrafish revealing a more general defect in CIII, extended co-IP experiments) and increasing the clarity of Figures and Figure legends. With these improvements, my main concerns have been addressed.

The only point that has not been addressed is the suggested rescue experiment with BR expression in zebrafish embryos, yet given the overall convincing and strong data that the defect is indeed due to loss of BR the manuscript is already convincing in its current form.

One minor suggestion to incorporate in the discussion or text: The authors currently favor the idea that loss of BR leads to a defect in CIII assembly – yet it could also be that import of CIII subunits is impaired, which could in turn lead to a defect in assembly. Since the current data does not allow to distinguish between these two possibilities (and experiments to distinguish between these possibilities will go beyond the scope of this manuscript), I would suggest mentioning this explicitly as a possibility for the primary function of BR.

Thank you for this suggestion. We will indeed pursue the possibility of import defects in our next paper and have added the following sentence in the Discussion : "However, our data currently cannot exclude the possibility that BR mediates CIII biogenesis by facilitating the import of its constituent subunits or their stability following translation/import, which will require further investigation."

Reviewer #3 (Remarks to the Author):

The authors have adequately addressed all comments raised by the reviewer and the revised manuscript is appropriate for publication.

Thank you for the recommendation.